
# 1 Improving the representation of HONO chemistry in
# 2 CMAQ and examining its impact on haze over China

Shuping Zhang,[1,2,3] Golam Sarwar,[4] Jia Xing,[2] Biwu Chu,[1,3,5] Chaoyang Xue,[1,3]
Arunachalam Sarav,[6] Dian Ding,[2] Haotian Zheng,[2] Yujing Mu,[1,3,5] Fengkui Duan,[2]
Tao Ma,[2] Hong He[1,3,5]
[1]State Key Joint Laboratory of Environment Simulation and Pollution Control, Research Center for
Eco-Environmental Sciences, Chinese Academy of Sciences, Beijing 100085, China
[2]State Key Joint Laboratory of Environment Simulation and Pollution Control, School of Environment,
Tsinghua University, Beijing 100084, China
[3]University of Chinese Academy of Sciences, Beijing 100049, China
[4]Center for Environmental Measurement and Modeling, U.S. Environmental Protection Agency, 109
T.W. Alexander Drive, Research Triangle Park, NC, 27711, USA
[5]Center for Excellence in Regional Atmospheric Environment, Institute of Urban Environment,
Chinese Academy of Sciences, Xiamen 361021, China
[6]Institute for the Environment, The University of North Carolina at Chapel Hill, 100 Eurpoa Drive,
Chapel Hill, NC 27514, USA
*Correspondence to*: Golam Sarwar(Sarwar.Golam@epa.gov), Jia Xing(xingjia@tsinghua.edu.cn), Hong
He(honghe@rcees.ac.cn)
**Abstract.**We compare Community Multiscale Air Quality (CMAQ) model predictions with measured
nitrous acid (HONO) concentrations in Beijing, China for December 2015. The model with the existing
HONO chemistry in CMAQ severely under-estimates the observed HONO concentrations with a
normalized mean bias of -97%. We revise the HONO chemistry in the model by implementing six
additional heterogeneous reactions in the model: reaction of nitrogen dioxide ($NO_2$) on ground surfaces,
reaction of $NO_2$ on aerosol surfaces, reaction of $NO_2$ on soot surfaces, photolysis of aerosol nitrate, nitric
acid displacement reaction, and hydrochloric acid displacement reaction. The model with the revised
chemistry substantially increases HONO predictions and improves the comparison with observed data
with a normalized mean bias of -5%. The photolysis of HONO enhances day-time hydroxyl radical by
almost a factor of two. The enhanced hydroxyl radical concentrations compare favorably with observed
data and produce additional sulfate via the reaction with sulfur dioxide, aerosol nitrate via the reaction
with nitrogen dioxide, and secondary organic aerosols via the reactions with volatile organic compounds.
The additional sulfate stemming from revised HONO chemistry improves the comparison with observed
concentration; however, it does not close the gap between model prediction and the observation during
polluted days.

## 34 1 Introduction

China has been suffering from haze pollution (Lelieveld et al., 2015) in which secondary particles
contribute more than 70% to the haze formation (Guo et al., 2014; Huang et al., 2014; Quan et al., 2014;
Zheng et al., 2015). However, the mechanism for the formation of high levels of secondary particles is



not yet clearly understood and most current air quality models tend to under-estimate particle
concentrations compared with observed data in China. Several secondary particle formation pathways
have been proposed, such as ①sulfate ($SO_4^{2-}$) formation via the heterogeneous oxidation of sulfur
dioxide ($SO_2$) promoted by hydrogen peroxide ($H_2O_2$) and/or ②nitrogen dioxide ($NO_2$) on mineral
dust (He et al., 2014; Huang et al., 2015; Ye et al., 2018), ③aqueous-phase oxidation of $SO_2$ promoted
by $NO_2$ in particle-bound water film (Wang et al., 2016; Li et al., 2017), ④aqueous-phase oxidation of
$SO_2$ by nitrous acid (HONO) produced from the photolysis of aerosol nitrate ($NO_3^-$) in particle-bound
water (Wang et al., 2016; Li et al., 2017), and ⑤$NO_3^-$ formation via efficient hydrolysis of dinitrogen
pentoxide ($N_2O_5$) on aerosol surfaces (Wang et al., 2017; Kulmala, 2018). However, the gap between
the model predictions and observed $SO_4^{2-}$ is persistent and still large (Zhang et al., 2019c).
Previous studies suggested that the underestimation of atmospheric oxidation capacity during haze
limited the formation of secondary particles (Sun et al., 2013; Gen et al., 2019; Tsona and Du, 2019).
As a hydroxyl radical (OH) source, HONO plays an important role in the oxidation of precursors (Stutz
et al., 2002; Kleffmann et al., 2005). However, the large underestimation of HONO (up to the ppb level)
is prevalent during haze simulations around the world (Li et al., 2012; Fu et al., 2019; Zhang et al.,
2019d). Moreover, HONO underestimation is reported to be highly related to the formation of fine
particulate matter ($PM_{2.5}$) (Wang et al., 2015; Xue et al., 2020), particularly for secondary $PM_{2.5}$.
Compared with summer, HONO concentrations in winter tend to be high when secondary particle
underestimation occurs (Li et al., 2018a; Zhang et al., 2019b). The underestimation of HONO may
partly explain the phenomenon of insufficient oxidant for the formation of secondary particles during
the winter haze (Li et al., 2018b; Li et al., 2018c).
Sarwar et al. (2008) compared the CMAQ predictions with HONO concentrations measured in
Philadelphia, PA, USA, during a summer month (July 2001) and reported that the model with only
gas-phase chemistry seriously under-estimates observed concentrations. They implemented HONO
emissions from motor vehicles, the heterogeneous reaction on the ground and aerosol surfaces, and the
photolysis of nitric acid ($HNO_3$) deposited on environmental surfaces, which improved predicted
HONO concentrations; however, the underprediction persisted. The model with the revised chemistry
enhanced OH and ozone ($O_3$) concentrations. Li et al. (2010) examined the impact of HONO chemistry
in Mexico City using the Weather Research and Forecasting model, coupled with chemistry
(WRF-CHEM). They added five different HONO reactions: ① the gas-phase reaction between NO
(nitric oxide) and OH, ②the heterogeneous reaction of $NO_2$ on the aerosol surfaces, ③the
heterogeneous reaction of $NO_2$ on the ground surfaces, ④the heterogeneous reaction of $NO_2$ with
semi-volatile organics, and ⑤$NO_2$ reaction with freshly emitted soot. The model successfully
reproduced observed HONO concentrations in Mexico City during March 2006. The model with the
HONO chemistry increased OH, $HO_2$ (hydroperoxyl radical), $O_3$, secondary organic aerosols (SOA),
$NO_3^-$, and ammonium ($NH_4^+$) and improved the comparison with observed data. The enhancements
were particularly high in the morning. However, the impact on $SO_4^{2-}$ was negligible. Czader et al. (2012)
compared CMAQv5.3 predictions with HONO measured during August and September 2006 in
Houston, TX, USA, and also reported that the model with gas-phase alone was not sufficient to explain
the observed data and predicted concentrations. They added HONO emissions, $NO_2$ hydrolysis, active
$NO_2$ chemistry, and conversion of $NO_2$ into HONO on organic materials covered surfaces, which
improved model performance for HONO and, subsequently, increased OH and $O_3$ concentrations.



83 Fu et al. (2019) studied a 5-day episode (January 4-8, 2017) in the Pearl River Delta of China during
84 which high levels of particles, $O_3$, and HONO concentrations were measured. They implemented four
85 additional reactions for HONO production into the model: ① relative humidity-dependent
86 heterogeneous reaction of $NO_2$, ②light-dependent heterogeneous reaction of $NO_2$, ③photolysis of
87 $NO_3^-$ , and ④photolysis of $HNO_3$ on surfaces. The model with the additional chemistry successfully
88 reproduced measured HONO concentrations which subsequently enhanced and improved $O_3$ and $PM_{2.5}$
89 predictions. Xing et al. (2019) examined the impact of HONO chemistry on SOA in the
90 Beijing-Tianjin–Hebei area (BTH) of China using the WRF-CHEM model during January 9-26, 2014.
91 They employed the homogeneous and heterogeneous HONO chemistry of Li et al. (2010) and reported
92 that the HONO chemistry could increase the average SOA concentration by ~46%. Zhang et al. (2019a)
93 employed the WRF-CHEM model to examine the impact of HONO chemistry on OH, $HO_2$, and SOA
94 concentrations in the BTH region during a winter haze period (November 29–Dec. 3, 2017). They
95 employed six HONO sources in the model: ① traffic emissions, ② soil emissions, ③ biomass
96 burning emissions, ④indoor emissions, ⑤heterogeneous reaction on aerosol surfaces, and ⑥
97 heterogeneous reaction on ground surfaces. The model reproduced observed HONO concentrations and
98 substantially elevated OH, $HO_2$, and SOA concentrations.

100 In this study, we employ the Community Multiscale Air Quality (CMAQ) model to simulate and
101 compare HONO predictions with observed data from the field campaign in Beijing. The field campaign
102 was conducted during December 7-22, 2015, in Beijing, China, during which high concentrations of
103 HONO and aerosols were measured.

104 **2 Methodology**

105 **2.1 Modeling framework and homogeneous HONO chemistry**

106 The Community Multiscale Air Quality (CMAQv5.3) (USEPA, 2019) (https://www.epa.gov/cmaq) was
107 used widely in this study. CMAQv5.3 includes the representation of important atmospheric processes
108 and has been used widely in air quality studies in many countries, including China (Byun and Schere,
109 2006; Sarwar et al., 2008; Xing et al., 2015). The modeling domain, which covered China and
110 consisted of $182 \times 232$ horizontal grid-cells with a $27 \times 27$ km horizontal resolution and 14 vertical
111 layers encompassing surface to 100 hPa. The first layer height of the model was about 36 m. The static
112 initial and boundary conditions from CMAQv5.3 were used for the study. A 22-day model spin-up
113 period was used to minimize the effect of initial conditions on model predictions. The Carbon Bond 6
114 (version 3, CB6r3) (Emery et al., 2015) chemical mechanism that contain six homogeneous reactions
115 related to HONO (Table 1) was used without any modification. CMAQv5.3 contains a treatment of
116 heterogeneous conversion of $NO_2$ at aerosol and ground surfaces (Sarwar et al., 2008), in which uptake
117 coefficient at aerosol surfaces and aera density of ground surfaces were revised in this study (Section
118 2.2). CMAQv5.3 accounts for HONO emissions from motor vehicles as $0.008 \times NO_x$ emissions which
119 were kept the same ($NO_X$ = oxides of nitrogen, $NO+NO_2$). Photolysis rates ($min^{-1}$) in CMAQv5.3
120 (J-values) are computed for photo dissociation reactions by Eq. (1). Absorption cross-section and
121 quantum yield data suggested by the International Union of Pure and Applied Chemistry (IUPAC) are
122 used for calculating photolysis rates of HONO (Table 1)


(http://iupac.pole-ether.fr/htdocs/datasheets/pdf). Absorption cross-section and quantum yield data
suggested by the IUPAC for NTR (organic nitrate) are used for calculating photolysis rates of CRON
(nitro-cresol) (Table 1) (http://iupac.pole-ether.fr/htdocs/datasheets/pdf).

$$J_i = \int_{\lambda 1}^{\lambda 2} F(\lambda)\sigma_i(\lambda)\phi_i(\lambda)d\lambda \qquad (1)$$
Note: $F(\lambda)$ is the actinic flux (photons cm$^{-2}$ min$^{-1}$ nm$^{-1}$), $\sigma_i(\lambda)$ the absorption cross section for the
molecule undergoing photo dissociation (cm$^2$ molecule$^{-1}$), $\phi_i(\lambda)$ the quantum yield of the photolysis
reaction (molecules photon$^{-1}$), and $\lambda$ the wavelength (nm).

We also instrumented the model with the Integrated Reaction Rate (IRR) option, which enabled
estimating the contribution of each reaction to the predicted HONO concentrations (Czader et al.,
2013). The Sulfur Tracking Model in CMAQv5.3 was used to quantitatively calculate the contribution
of each reaction to predicted $SO_4^{2-}$ concentration (Mathur et al., 2008).

**Table 1 Gas-phase chemical reactions related to HONO in CB6r3**

| Reaction Number | Reaction | Reaction Rate Constant (k) |
|---|---|---|
| 1 | NO + OH = HONO | $k = \left\{\dfrac{k_0[M]}{(1 + k_0[M]/k_1)}\right\} 0.81^{\left\{1+\left[log_{10}\left(\frac{k_0[M]}{k_1}\right)/0.87\right]^2\right\}^{-1}}$ <br><br> $k_0 = 7.4 \times 10^{-31}\left(\dfrac{T}{300}\right)^{-2.4}$ <br><br> $k_1 = 3.3 \times 10^{-11}\left(\dfrac{T}{300}\right)^{-0.3}$ |
| 2 | NO + NO₂ + H₂O = 2.0×HONO | $k = 5.0 \times 10^{-40}$ |
| 3 | HONO + HONO = NO + NO₂ | $k = 1.0 \times 10^{-20}$ |
| 4 | HONO = NO + OH | $J_{HONO}$ |
| 5 | HONO + OH = NO₂ | $k = 2.5 \times 10^{-12}e^{(260/T)}$ |
| 6 | CRON = HONO + HO2 + FORM + OPEN | $J_{NTR}$ |

Note: NO = nitric oxide, $NO_2$ = nitrogen dioxide, OH = hydroxyl radical, $HO_2$ = hydroperoxy radical,
$H_2O$ = water vapor, HONO = nitrous acid, CRON = nitro cresol; FORM = formaldehyde, OPEN =
aromatic ring open product, [M] = total pressure (molecules/cm$^3$), T = air temperature (K), and k = rate
constant. First-order rate constants are in units of s$^{-1}$, second-order rate constants are in units of cm$^3$
molecule$^{-1}$ s$^{-1}$, third-order constants are in units of cm$^6$ molecule$^{-2}$ s$^{-1}$. CMAQv5.3 converts
cm-molecule-s units into ppm-min units before solving the system of ordinary differential equations for
chemistry. $J_{HONO}$ = photolysis of HONO, and $J_{NTR}$ = photolysis of NTR (organic nitrate).






## 2.2 Heterogeneous HONO chemistry

The understanding of heterogeneous HONO chemical reactions and parameter method is evolving.
Investigators have proposed hydrolysis of $NO_2$ on the humid aerosol surfaces, heterogeneous
conversion of $NO_2$ on ground surfaces, photolysis of $NO_3^-$, catalytical formation on soot particles and
acid displacement process in the atmosphere during the past several years (Stemmler et al., 2006; Liu et
al., 2014; Karamchandani et al., 2015; VandenBoer et al., 2015; Tong et al., 2016; Ye et al., 2016; Ye et
al., 2017; Lu et al., 2018; Xu et al., 2018b; Gen et al., 2019; Zhang et al., 2019d). Xue et al. (2020) and
Liu et al. (2019) recently measured summertime atmospheric HONO concentrations in a rural area in
China and performed simulations using a box model with updated chemical reactions for HONO
production published in the literature. They reported that the simulations generally reproduced
observed HONO concentrations using the updated HONO chemical reactions. However, the box model
did not consider horizontal and vertical transportation, limiting the impact of HONO formation on air
quality. We implement these updated chemical reactions into a three-dimensional (3D) air quality
model, CMAQv5.3, to examine their impacts on air quality.

Hydrolysis processes on the humid aerosol surfaces is an important HONO-producing reaction in the
atmosphere (An et al., 2012; Cui et al., 2018). And we use the uptake coefficient (Table 2) employed by
Liu et al. (2019) at night-time (Reaction 7a). The reaction on aerosols can be enhanced by light (Zhang
et al., 2019b); thus, we use a radiation-dependent uptake coefficient during day-time (Reaction 7b). We
use CMAQv5.3-calculated aerosol surface area-to-volume ratio ($S/V_a$) to calculate rate constant for the
reaction on aerosol surfaces. Heterogeneous conversion of $NO_2$ on ground surfaces also has been
studied intensively in the laboratory and field (Li et al., 2018a). Vertical night-time profile
measurements suggest that heterogeneous HONO formation on the ground is the dominant reaction;
thus, we also use this reaction. Similar to the heterogeneous reaction on aerosol surfaces, we employ an
uptake coefficient used by Liu et al. (2019) for the reaction at night (Reaction 8a) and a
radiation-dependent uptake coefficient during day-time (Reaction 8b). Following the suggestions of Li
et al., (2019) and Liu et al., (2019) , we use a value of 1.7/H (H is the model's first-layer height) for
surface area-to-volume ratio of ground ($S/V_g$) to calculate the rate constant for the reaction on ground
surfaces.

Ye et al. (2016) proposed that the photolysis of $NO_3^-$ can lead to HONO production in the atmosphere
and reported that its photolysis rates can be several hundred times faster than the photolysis rates of
$HNO_3$. Bao et al. (2018) also reported similar photolysis rates of $NO_3^-$. Fu et al. (2019) used this high
photolysis rate in their study to examine the winter-time HONO production in Hong Kong. However,
Romer et al. (2018) reported that such high photolysis rates of $NO_3^-$ are not consistent with observed
data over the Yellow Sea and should not be used in air quality models. They suggested that the
photolysis rates of $NO_3^-$ in air quality models should be 1 to 30 times the photolysis rate of $HNO_3$. For
photolysis of $NO_3^-$ , we use a photolysis rate of 30 times the photolysis rate of $HNO_3$ (Reaction 9).

HONO formation on soot particles can be catalytically enhanced in the presence of artificial solar
radiation and lead to persistent reactivity of soot over long periods (Monge et al., 2010). The surface of
soot particles as a heterogeneous conversion media has been reported by several studies (Monge et al.,



2010; Liu et al., 2014; Spataro and Ianniello, 2014; Cui et al., 2018). The reported heterogeneous
uptake coefficient on soot ranges from $10^{-8}$ to $10^{-6}$, with HONO yields ranging between 50% and 100%
(Spataro et al., 2013). This heterogeneous soot photochemistry potentially may contribute to day-time
HONO concentration. We also employ the reaction using the upper limit of the reported uptake
coefficient and calculate the HONO formation rate following Spataro et al. (2013) (Reaction 10).

VandenBoer et al. (2013) reported that deposited HONO can react with carbonates or soil at night and,
subsequently, be released from the soil during the day by reactions with atmospheric $HNO_3$ and HCl
(hydrochloric acid). They suggest that this acid displacement process can contribute to a substantial
fraction of day-time HONO. We also use this process (Reactions 11 and 12) and employ a parameter
similar to that of Liu et al. (2019), except that we utilize a displacement efficiency of 6% for $HNO_3$ and
20% for HCl following VandenBoer et al. (2015).

Zhou et al., (2003) reported that $HNO_3$ deposited on environmental surfaces can undergo rapid
photolysis leading to day-time HONO production. Several studies (Sarwar et al., 2008; Fu et al., 2019;
Liu et al., 2019) included such a reaction in their models. However, we do not include it because the
rate constant has high uncertainty and it could also pose a problem for performing long-term model
simulations. For long-term (annual and multiyear) that the deposited amount of $HNO_3$ could
accumulate with time, which could continue increasing the HONO production rates with time. Soil can
emit HONO and other nitrogen-containing compounds (Su et al., 2011; Oswald et al., 2013). Rasool et
al. (2019) implemented these emissions into CMAQv5.3 by using a mechanistic representation of the
underlying processes and examined their impacts on air quality over North America. The impacts of
HONO emitted from soil are generally low, and we do not include these emissions in this study.

Table 2 Heterogeneous HONO reactions used in this study

| Reaction No. | Reaction | Reaction Rate Constant (k) | Uptake coefficient ($\gamma$) | Reference |
|---|---|---|---|---|
| 7a. | $NO_2$ + aerosol = 0.5×HONO + 0.5×$HNO_3$ | $k = \frac{1}{4}\gamma v_{NO2}\frac{S}{V_a}$ | $\gamma_{an} = 8\times10^{-6}$ | (Liu et al., 2019) |
| 7b. | $NO_2$ + aerosol + hv = 0.5×HONO + 0.5×$HNO_3$ | $k = \frac{1}{4}\gamma v_{NO2}\frac{S}{V_a} \times \frac{J}{J_{max}}$ | $\gamma_{ad} = 1\times10^{-3}$ | (Liu et al., 2019) |
| 8a. | $NO_2$ + ground = HONO | $k = \frac{1}{8}\gamma v_{NO2}\frac{S}{V_g}$ | $\gamma_{gn} = 4\times10^{-6}$ | (Li et al., 2018a; Liu et al., 2019) |
| 8b. | $NO_2$ + ground + hv = HONO | $k = \frac{1}{8}\gamma v_{NO2}\frac{S}{V_g} \times \frac{J}{J_{max}}$ | $\gamma_{gd} = 6\times10^{-5}$ | (Liu et al., 2019) |
| 9 | $NO_3^-$ + hv = 0.67×HONO + 0.33×$NO_2$ | $J = 30\times J_{HNO3}$ | | (Romer et al., 2018) |
| 10 | $NO_2$ + EC= 0.61×HONO + 0.39×NO | $k = \frac{1}{4}\gamma v_{NO2}\frac{S\_BET}{V}$ | $\gamma = 2\times10^{-6}$ | (Spataro and Ianniello, 2014) |



| 11 | HNO$_3$ + NaNO$_2$ (s) = HONO + NaNO$_3$ (s) | k=0.06V$_{dep\_HNO3}$/H | | (VandenBoer et al., 2015) |
| 12 | HCl + NaNO$_2$ (s) = HONO + NaCl (s) | k=0.2V$_{dep\_HCl}$/H | | (VandenBoer et al., 2015) |

Note:
k = first order rate constant (sec$^{-1}$), $\gamma$ = heterogeneous uptake coefficient (-), $\gamma_{an}$ = night-time
heterogeneous uptake coefficient on aerosol, $\gamma_{ad}$ = day-time heterogeneous uptake coefficient on aerosol,
$\gamma_{gn}$ = night-time heterogeneous uptake coefficient on ground, $\gamma_{gd}$ = day-time heterogeneous uptake
coefficient on ground, S/V$_a$= density of aerosol surface; S/V$_g$= density of ground surface; $v$ = mean
molecular speed (m/s), HNO$_3$ = nitric acid, NaNO$_2$ = sodium nitrite, NaCl = sodium chloride, J = NO$_2$
photolysis rate, J$_{max}$ = maximum NO$_2$ photolysis rate, V$_{dep\_HNO3}$ = deposition velocity of HNO$_3$ (m/s),
V$_{HCl}$ = deposition velocity of HCl (m/s), H = the first-layer height (m), and S_BET/V = BET surface
area-to-volume ratio that we calculate as follows: CMAQv5.3 predicted elemental carbon (EC) ($\mu$g/m$^3$)
$\times$ 1.0 $\times 10^{-6}$ (g/$\mu$g) $\times$122 m$^2$/g, NO$_3^-$ = aerosol nitrate, EC= elemental carbon. Reactions 7a, 7b, 8a, and
8b are revised from CMAQv5.3, while reaction numbers 9 ,10, 11 and 12 are newly added reactions.
**2.3 Simulation cases**
We performed two different simulations using CMAQv5.3 for December 7-22, 2015. One simulation
denoted by "ORI" used the gas-phase HONO chemistry in CB6r3 along with the existing
heterogeneous hydrolysis of NO$_2$ in CMAQv5.3. The other simulation denoted by "REV" used the
gas-phase HONO chemistry in CB6r3 and the heterogeneous reactions presented in Table 2. For this
simulation, we removed the existing heterogeneous hydrolysis of NO$_2$ in CMAQv5.3. Both simulations
used the same HONO emissions (section 2.1). We also completed several additional sensitivity
simulations as discussed in Section 3.0.

We used the ABaCAS national emissions inventory (http://www.abacas-dss.com) which resulted in
great performance in simulating both NO$_2$ and fine particle (PM$_{2.5}$).   In previous studies, Zhao et al.
(2018) and Zheng et al. (2019) used these emissions and reported a normalized mean bias (NMB) of
4% for NO$_2$ and -17% for PM$_{2.5}$. Meteorological fields for CMAQv5.3 were simulated using the
Weather Research and Forecasting  model version 3.8 (WRFv3.8) (Skamarock and Klemp, 2008).
WRF has consistent parameterization for cloud fraction simulation, as well as other climate models
(see (Xu and Krueger, 1991) and (Xu and Randall, 1996) for a review on this topics). We compared
WRF predictions with observed temperature, wind speed, and water vapor mixing ratio in China (Fig.
S1). Mean bias (MB) and root mean square error (RMSE) for temperature, wind speed and MB for
water vapor mixing ratio meet the benchmark limits suggested by Emery et al.(2001) (Table S1).
**2.4 Observation data**
A field campaign was conducted during December 7-22, 2015, at the Research Center for
Eco-Environmental Sciences (40.01° N, 116.35° E) to measure atmospheric pollutants and
meteorological parameters. Atmospheric concentrations of HONO were measured using a stripping coil
(SC) equipped with ion chromatograph (IC). The details of the instrument have been described
elsewhere (Xue et al., 2019a). We also completed a statistical analysis of the measurements from the
instrument with data from three other methods and concluded that it can provide reliable measurements.





The instrument has a minimum detection limit of 4.0 ppt and has been used in several field campaigns
(Xue et al., 2019). The concentrations of $NO_2$ and $NO_x$ were measured by a nitrogen oxide analyzer
(Thermo 42i, Thermo Fisher, USA). Sulfur dioxide ($SO_2$) was measured by a pulsed fluorescence
analyzer (Thermo 43i, Thermo Fisher, USA). Fine particles ($PM_{2.5}$) was measured using an Aerosol
Monitor (TSI, Thermo Fisher TEOM 1405). Relative humidity (RH), temperature, wind speed (WS),
wind direction (WD), and other metrological data were measured by an automatic weather-monitoring
system. Daily atmospheric $SO_4^{2-}$ and $NO_3^-$ samples were collected on the roof of a three-story building
on the campus of Tsinghua University in Beijing (40.0° N, 116.3° E) and measured by ion
chromatography. Details of the measurements method are described by Ma et al. (2020). The hourly
averaged concentrations of the main chemical species of $PM_{2.5}$ were measured by the Gas and Aerosol
Compositions Monitor (IGAC, Fortelice International Co., Taiwan) monitoring system (Feng et al.,
2018). The observed vertical HONO concentrations from the study of Meng et al.(2020) was measured
in December of 2016. The OH measurements in January of 2014 were achieved from the study of Tan
et al.(2018).

**3  Results and discussions**
**3.1 Comparison of model prediction with observed HONO**
Observed HONO concentrations vary with time, range between 0.04 ppb to 8 ppb, and contains 11
episodes in which the daily peak concentration exceeds 3.3 ppb (Fig. 1a). The high HONO
concentration occurs during low wind speeds (Fig. S1). The average HONO concentration during the
period is comparable to the reported values for other cities (Table S2). Predicted HONO concentrations
obtained with the ORI case are substantially lower than the observed data. In contrast, predicted HONO
concentrations obtained with the REV case are substantially higher than those obtained with the ORI
case and generally similar to the observed data at night. The ORI case misses the peak values for all
episodes, whereas the REV case captures peak values for most episodes. The observed average
concentration during the measurement period is 2.5 ppb, the ORI case only predicts an average
concentration of 0.1 ppb, whereas the REV case predicts an average concentration of 2.3 ppb. The
NMB of HONO is reduced from -96.5% with the ORI case to -4.8% with the REV case.

Consistent with observations at other cities (Bernard et al., 2015; Fu et al., 2019), the diurnal variation
of observed HONO concentrations in Beijing also reveals higher night-time concentrations than
day-time values (Fig. 1b). The predictions with ORI case are an order of magnitude lower than the
observed diurnal concentrations. The diurnal variation with the REV case shows a remarkable
enhancement of night-time HONO concentrations to levels similar to the observed concentrations. It
also increases day-time concentrations, however, predicted values are substantially lower than the
observed data, which suggests that additional sources are needed to close the gap between observed
and predicted day-time HONO concentrations. Night-time and day-time heterogeneous reaction and
other updated reactions contribute to the improvement of HONO diurnal pattern. More detailed
analysis about this great enhancement is included in section 3.2. The diurnal pattern of the predicted
HONO concentrations with the REV agrees better with the observed diurnal pattern.

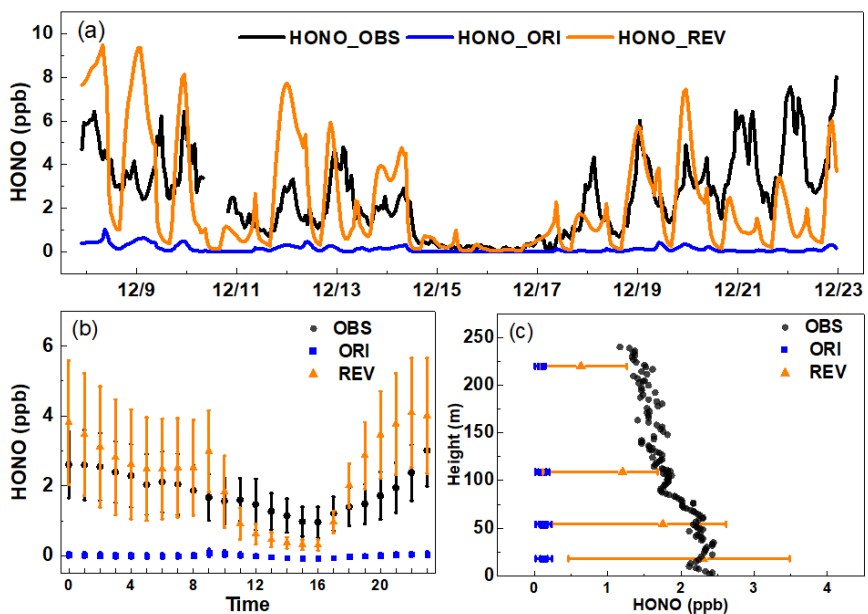

**Fig. 1 A comparison of simulated and observed HONO concentrations in Beijing (a) time series (b) diurnal variation, and (c) vertical comparison.**

We compare predicted vertical distribution with observed vertical HONO concentrations (39.97° N, 116.38° E) from the study of Meng et al.(2020) (Fig. 1c). The measured concentration is the highest at the surface (2.3 ppb), and concentrations decrease with increasing altitude to a value of ~1.2 ppb at ~200 m, which supports the dominant role of the surface HONO production. Predicted HONO levels with ORI case are too small, whereas predictions with the REV agree better with observed data not only at the surface but also aloft, which provides validity of the simulation results. Consistent with previous HONO vertical concentrations and flux measurements (VandenBoer et al., 2013; Li et al., 2018a), HONO concentration at the surface layer is highest. Model simulated HONO concentrations (Fig. 1c) show a decreasing trend with height similar to the trend in observation data reported by Meng et al.(2020). Model HONO concentrations at upper layers (above 50 m in Fig. 1c) are slightly under-estimated. Model HONO concentrations in these layers are produced mainly by the heterogeneous reaction of $NO_2$ on aerosol surfaces and the reaction of $NO+OH$. Aerosol indirect effects can reduce photolysis rate of HONO (Xing et al., 2017). Decreasing photolysis can improve HONO concentrations in the upper layers in polluted air.

The $HONO/NO_2$ ratio is used as an indicator to estimate the efficiency of heterogeneous $NO_2$–HONO conversion (Kleffmann et al., 2006; Li et al., 2012). The observed $HONO/NO_2$ ratios ranging between 0.003 and 0.15 are much higher than reported values in the vehicle exhausts which suggests that HONO formation is governed mainly by the secondary production (Kirchstetter et al., 1996; Kurtenbach et al., 2001). The diurnal variation of observed and predicted $HONO/NO_2$ ratios are shown in Fig. S2. The predicted $HONO/NO_2$ ratios increase substantially with REV compared with the ORI case. The average ratio of $HONO/NO_2$ increases from 0.0027 with ORI and to 0.053 with REV, which





is in agreement with the observed value of 0.055. The NMB of hourly average simulated HONO/NO₂
ratios at night-time decreases from -94.4% with ORI and to -34.2% with REV. The model results
suggest that NO₂ heterogeneous conversion is the most important reaction for simulating atmospheric
HONO concentrations.

According to our detailed literature review in methodology part, uncertainties of HONO prediction
might be largely associated with four key parameters and inputs including the uptake coefficient of
NO₂ at ground surface, the aerosol nitrate photolysis rates, the daytime photolysis rate, as well as the
baseline NOₓ emissions. Sensitivity analysis was conducted to examine the influences from those
parameters, suggesting that HONO concentration could be doubled with different parameters (see
Supplementary Information). Besides, some sources including the photolysis of deposited HNO₃, soil
emission and traffic emission could also affect predicted HONO concentration, while the importance of
these sources is difficult to quantify. Future studies in improving the accuracy of these parameters are
important to reduce the uncertainties of HONO prediction.
**3.2 Relative contribution of different HONO reactions**
To gain insights into HONO reactions, production rates of different reactions are calculated, and the
diurnal variation of the production rates is presented in Fig. S5. The production rates from the
heterogeneous reaction on ground surfaces (denoted HONOfrNO2G) are higher during the day than
those at night because of the higher rate constant. During night-time (18-5 h / 6:00 p.m.-5:00 a.m.), it
dominates the HONO production with an average production rate of 1.4 ppb/h. Similar to
HONOfrNO2G, the production rates from the heterogeneous reaction on aerosol surfaces (denoted
HONOfrNO2A) are also higher during day-time compared with those at night-time. It contributes an
average production rate of 0.2 ppb/h during night-time. The contribution of other reactions to
night-time HONO production are relatively small (<0.03ppb/h). During day-time (6-17 h / 6:00
a.m.–5:00 p.m.), HONOfrNO2G also dominates the production with an average contribution of 2.05
ppb/h. HONOfrNO2A is the second most important contributor during day-time with an average
production rate of 0.54 ppb/h. The photolysis of NO₃⁻ is the third contributor with an average
production of 0.04 ppb/h. Gas-phase reactions collectively contribute an average production rate of
~0.41 ppb/h. The NO+OH reaction is the most important gas-phase reaction, producing HONO at an
average rate of 0.37 ppb/h. The average day-time production rates of the acid displacement reactions of
HNO₃ and HCl are 0.25 ppb/h and 0.03 ppb/h, respectively. The contribution of the reaction on
elemental carbon (EC) is even smaller (<0.01 ppb/h). Day-time production from the heterogeneous
reaction on ground and aerosol surfaces is greater than the combined production from all other
reactions. Although updated day-time reaction rates are higher than that of night-time, accelerated
photochemical loss slow down the HONO increase during day-time.

The relative contribution of the chemistry updates to HONO formation (REV) is shown in Fig. 2.
HONOfrNO2G is the most important reaction, contributing ~86.2% of night-time HONO production.
HONOfrNO2A is the second largest contributor, representing ~12.3% of night-time HONO production.
During day-time, HONOfrNO2G contributes ~64.7% of the HONO production, whereas
HONOfrNO2A is the second largest contributor, representing 12.6% of the HONO production.
Day-time HONO production rate from HONOfrNO2A is higher than that at night-time due to the
higher rate constant. Consequently, the relative importance of day-time heterogeneous reaction on





aerosol surfaces increases, whereas the relative importance of day-time heterogeneous reaction on
ground surfaces decreases. The acid displacement reaction of $HNO_3$ contributes 11% to day-time
HONO formation, and its contribution peaks at 5 p.m. (17 h). The average contribution of gas-phase
reactions, photolysis of $NO_3^-$ and acid displacement reactions to day-time HONO production are 9.4%,
1.0%, and 1.3%, respectively. Note that the reaction of OH+NO becomes important in the morning (9
to 10 a.m.) during which it contributes 17.4% of the total HONO production. Averaged over the day
and night, HONOfrNO2G is the most significant reaction, contributing 75.6% of the HONO production.
HONOfrNO2A is the second largest contributor, representing 12.3% of the HONO production. The
gas-phase reactions and the acid displacement reaction of $HNO_3$ are the third most important
contributor each accounting for 5.6% of HONO production. Although HONOfrNO2G had a relatively
lower uptake coefficient than the aerosol surface reaction, the reaction rate was large because of the
greater ground surface area density (0.047 $m^2$ $m^{-3}$) compared with the aerosol surface area density
(0.0014 $m^2$ $m^{-3}$).

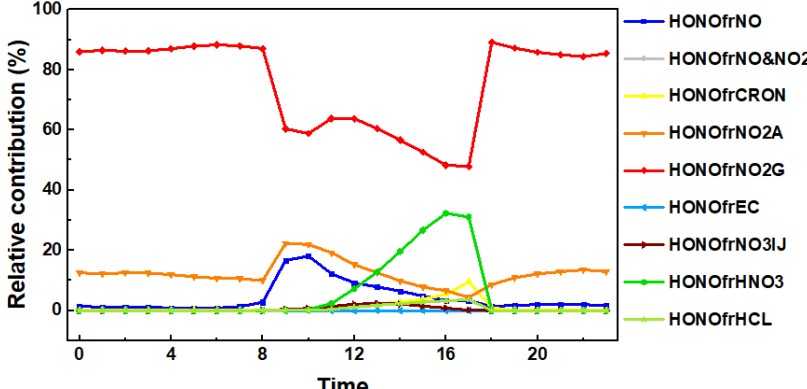


**Fig. 2 Relative contribution of different HONO reactions to near-ground-level HONO concentration in**
**Beijing in December. The production from the NO+OH reaction is denoted as HONOfrNO, the production**
**from the NO+NO2+H2O reaction is denoted as HONOfrNO&NO2, the production from cresol is denoted as**
**HONOfrCRON, the production from the heterogeneous reaction on ground surfaces is denoted as**
**HONOfrNO2G, the production from the heterogeneous reaction on aerosol surfaces is denoted as**
**HONOfrNO2A, the production from the reaction of EC is denoted as HONOfrEC, the production from the**
**photolysis of NO3- is denoted as HONOfrNO3IJ, the production from the acid displacement reaction of HNO3**
**is denoted as HONOfrHNO3, and the production from acid displacement reaction of HCl is denoted as**
**HONOfrHCL.**

**3.3 Impacts of HONO chemistry on hydroxyl radical concentration**

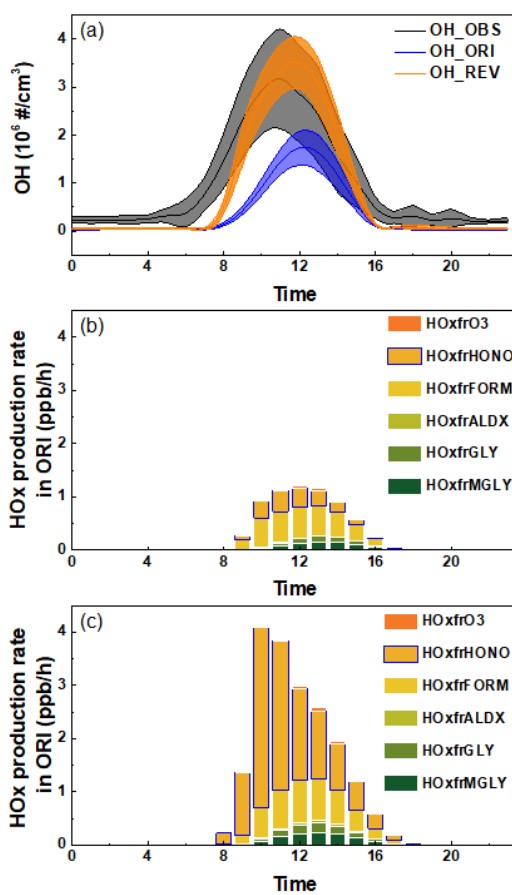


**Fig 3 (a) A comparison of simulated and observed diurnal variation of OH. Shadow in Fig. 3a indicates the**
**range of observation. (b) HO$_x$ formation rates from different photolytic reactions with the ORI case and (c)**
**HO$_x$ formation rates from different photolytic reactions with the REV case. The production of HO$_x$ from the**
**O$_3$ photolysis is denoted as HOxfrO3, the production of HO$_x$ from the HONO photolysis is denoted as**
**HOxfrHONO, the production of HO$_x$ from the formaldehyde photolysis is denoted as HOxfrFORM, the**
**production of HO$_x$ from the higher aldehyde photolysis is denoted as HOxfrALDX, the production of HO$_x$**
**from the glyoxal photolysis is denoted as HOxfrGLY, and the production of HO$_x$ from the methyl glyoxal**
**photolysis is denoted as HOxfrMGLY.**

Enhanced HONO production increases model OH concentration via photolysis. We compare predicted
OH concentrations with observed winter data (40.41° N, 116.68 ° E) reported by Tan et al. (2018) in
Fig 3a. Observed concentrations are low (~2-3×10$^5$ /#/cm$^3$) at night and rapidly increase in the morning
reaching a peak value of ~3×10$^6$ /#/cm$^3$ at around 11:00 a.m., then slowly decrease to the low nightly
values. The ORI case under-predicts the observed peak value by a factor of ~2, and the model peak
time occurs 1 to 1.5 hours after the observed peak time, which is consistent with a previous study in





which additional HONO reactions increased OH levels by a factor of >2 (Xue et al., 2020). In addition,
the morning enhancement rate with ORI is very low compared with the observed rate. In contrast, the
REV case reproduces the observed peak and improves the timing of the peak. The morning
enhancement rate also substantially increases and closely tracks the observed enhancement rate. The
daily average concentration of OH with REV increases by ~98% compared with that obtained with
ORI. Thus, the REV case successfully captures the morning enhancement rate and the peak, and
improves the timing of the peak in observed OH data in Beijing. Overall, it captures the observed OH
concentration in Beijing much better than the model with the original chemistry. To examine the
vertical extent of the impact on OH, predicted OH concentrations with altitude are shown in Fig. S6
(40.0 ° N, 116.3 ° E). Predicted OH concentration with ORI is the lowest at the surface and increases
with altitude primarily because of higher $O_3$ aloft. The REV case increases OH concentration not only
due to the surface HONO but also aloft. However, the impact on OH decreases with altitude as the
HONO production decreases with altitude.
Various photolytic reactions, including the photolysis of $O_3$, HONO, formaldehyde, higher aldehyde,
glyoxal, and methyl glyoxal, produce $HO_x$ ($OH+HO_2$) are in the model. To understand the relative
impacts of these HONO reactions on $HO_x$ production, we compare the diurnal production rates of $HO_x$
from these reactions in Fig. 3b and c. In the ORI case (Fig. 3b), the production of $HO_x$ is relatively
small and dominated by the photolysis of HONO and formaldehyde. The photolysis of HONO and
formaldehyde start producing $HO_x$ at 9 a.m. which initiates day-time atmospheric chemistry. From late
morning, the production of $HO_x$ from glyoxal and methyl glyoxal also contributes to the continuation
of day-time atmospheric chemistry. In our simulation, glyoxal and methyl glyoxal originate from the
oxidation of aromatics in the atmosphere because isoprene concentration in Beijing is low in winter.
Averaged over the entire day, the photolysis of formaldehyde is the largest contributor (0.14 ppb/h) and
the photolysis of HONO is the second largest contributor (0.08 ppb/h) to the total $HO_x$ production rate.
The production from $O_3$ and higher aldehyde photolysis are small as their concentrations are low.
In contrast, the $HO_x$ production rates in the REV case are much higher than those in the ORI case
because of the enhanced formation from HONO (Fig. 3c). The photolysis of HONO produces $HO_x$ in
the morning, which then kick-starts day-time atmospheric chemistry at 8 a.m. (1 h earlier than in the
ORI case) and continues to play an important role during the entire day. From late morning, the
production of $HO_x$ from formaldehyde, glyoxal, methyl glyoxal, and higher aldehyde also contributes
to the continuation of day-time atmospheric chemistry. The production of $HO_x$ from glyoxal, methyl
glyoxal, and higher aldehydes plays a larger role compared with that in the ORI case because of higher
concentrations produced by the enhanced oxidation of aromatics by higher OH. The photolysis of
HONO is the largest contributor (0.5 ppb/h) to the overall $HO_x$ production rate averaged over the entire
day while the photolysis of formaldehyde is the second largest contributor (0.18 ppb/h). Thus, HONO
plays a crucial role in producing OH in the morning, without updated reactions, the start of day-time
atmospheric chemistry is delayed; and the reaction rates are slower, it also plays an important role in
atmospheric chemistry throughout the day. Many other photolytic reactions also produce $HO_x$ in the
model; however, the productions from the other pathways are small and do not affect our calculation,
hence, they are not shown in the figure.
HONO can affect greatly the daily OH budget (Harris et al., 1982; Li et al., 2018c; Lu et al., 2019; Xue


et al., 2020). Our simulations with the additional HONO reactions enhances OH, which in turn
increases $HO_2$ by the fast conversion between OH and $HO_2$ radicals (Heard and Pilling, 2003; Lu et al.,
2012). The reaction rate of the $HO_2+NO$ reaction increases from 1.8 ppb/h in ORI to 3.6 ppb/h in REV.
This indicates that the HONO chemistry also indirectly promotes the formation of OH by increasing
the activity of $HO_2$. This highlights the promoting role of HONO in gas-phase radicals.
Increased OH concentration oxidizes additional volatile organic compounds (VOCs), lowers the
concentrations of precursor species, and increases the concentrations of secondary species (Table S3).
Enhanced oxidation of VOCs, sulfur dioxide, and $NO_2$ leads to secondary pollutants, including $SO_4^{2-}$,
$NO_3^-$, $NH_4^+$, and SOA, which are discussed in the next section.
**3.4 Impacts of HONO chemistry on the formation of secondary particles**
Daily averaged model predicted $SO_4^{2-}$, $NO_3^-$, and $NH_4^+$ concentrations are compared with observed
data in Beijing in Fig. 4. The ORI case captures the observed trend but generally under-estimates the
observed $SO_4^{2-}$ concentrations, whereas the REV case enhances $SO_4^{2-}$ concentrations and closes the gap
between model predictions and observation data. Over the entire simulation period, the average
concentration of $SO_4^{2-}$ is increased from 13.3 µg/m$^3$ to 15.8 µg/m$^3$ (19%). CMAQv5.3 includes six
chemical pathways for the conversion of $SO_2$ into $SO_4^{2-}$ (Sarwar et al., 2011). These are ① the
gas-phase oxidation of $SO_2$ by OH, aqueous-phase oxidation of S(IV) (the sum of $SO_2$•$H_2O$ [hydrated
$SO_2$], $HSO_3^-$ [bisulfite ion] and $SO_3^{2-}$ [sulfite ion]) by ② $H_2O_2$ (hydrogen peroxide), ③ $O_3$, ④ PAA
(peroxyacetic acid), ⑤ MHP (methylhydroperoxide), and ⑥ oxygen catalyzed by the iron (Fe[III])
and manganese (Mn[II]). We utilized the sulfate tracking model to examine $SO_4^{2-}$ production from
these chemical pathways over Beijing. The $SO_4^{2-}$ production from the gas-phase oxidation of $SO_2$ by
OH in the REV is ~79% greater than that of the ORI case because of the higher OH concentration from
HONO photolysis. $SO_4^{2-}$ production from the aqueous-phase oxidation of S(IV) by $H_2O_2$ in the ORI is
relatively small because the predicted $H_2O_2$ concentration is also small in winter. However, the REV
case enhances $H_2O_2$ concentration, which consequently also increases the $SO_4^{2-}$ production from this
pathway. The other chemical pathways produce similar concentrations in both models, except the
oxygen catalyzed by the Fe[III] and Mn[II] pathway, which produce slightly lower $SO_4^{2-}$ production in
the REV case because of the competition among different chemical pathways and greater oxidation by
the OH initiated pathway.
**Table 3. Predicted $SO_4^{2-}$ concentration in Beijing from different chemical pathways in CMAQv5.3**

| Chemical pathway | Average $SO_4^{2-}$ concentration in ORI (µg/m$^3$) | Average $SO_4^{2-}$ concentration in REV (µg/m$^3$) |
|---|---|---|
| $SO_2 + OH$ | 2.23 | 3.99 |
| $S(IV) + H_2O_2$ | 0.25 | 0.41 |
| $S(IV) + O_3$ | 0.02 | 0.02 |



| | | |
|---|---|---|
| S(IV) + O$_2$ (TMI) | 0.61 | 0.50 |
| S(IV) + MHP | 0.01 | 0.01 |
| S(IV) + PAA | <0.01 | <0.01 |


TMI: S(IV) oxidation by oxygen catalyzed by Fe[III] and Mn[II]

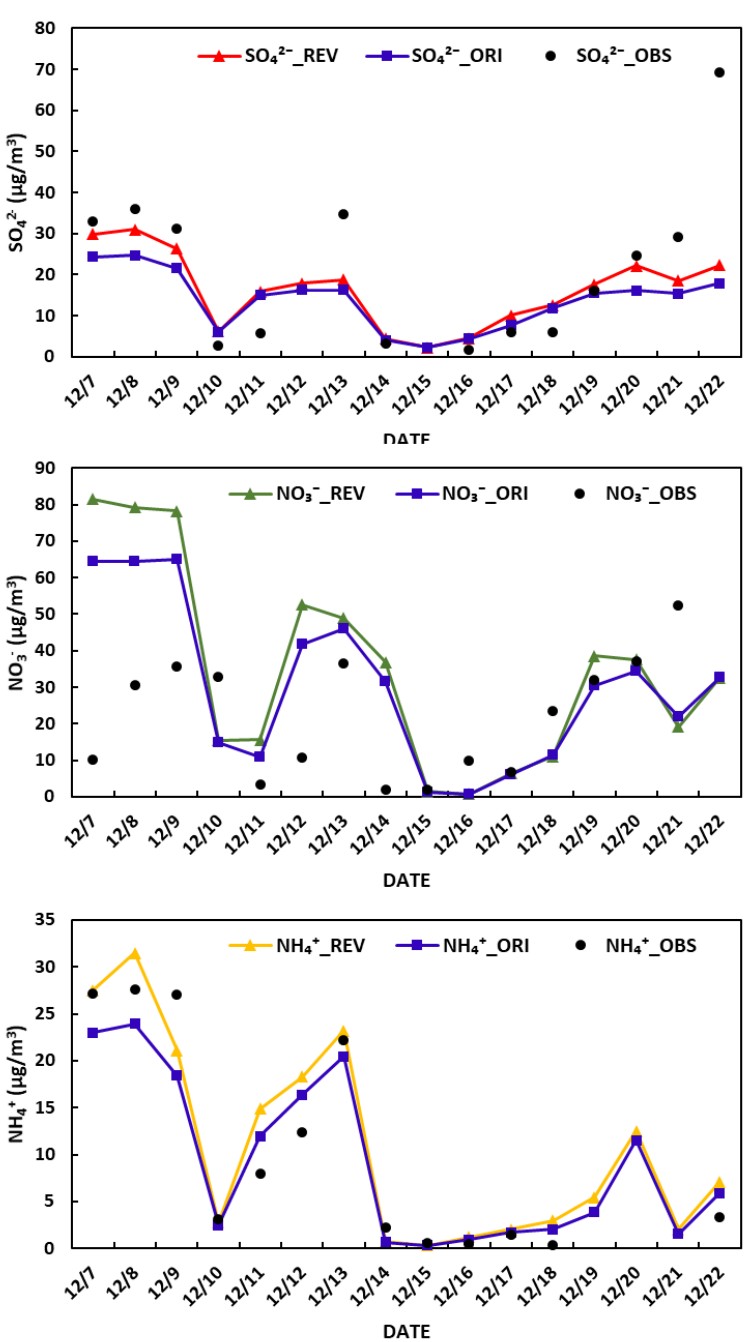

**Fig. 4 A comparison of simulated and observed daily averaged sulfate, nitrate and ammonium concentration in Beijing**

Additional $SO_4^{2-}$ production is needed in the model to close the gap between the model prediction and

observed data. Several investigators have proposed other pathways that can generate additional $SO_4^{2-}$
production. For example, Gen et al. (2019) conducted laboratory experiments and reported that the
photolysis of $NO_3^-$ can generate N(III) (HONO+$NO_2^-$) in aerosol liquid water, which oxidizes S(IV)
into $SO_4^{2-}$ . Zheng et al. (2020) recently incorporated such a pathway and reported that it can enhance
$SO_4^{2-}$ production and can explain 15% to 65% of the gap between model predictions and observed
$SO_4^{2-}$ concentrations in China. Shao et al. (2019) implemented several additional heterogeneous $SO_4^{2-}$
formation pathways for oxidation of S(IV) in aerosol liquid water and reported that the pathways can
enhance $SO_4^{2-}$ production by 20% in China. Wang et al. (2020) recently reported that S(IV) can be
oxidized by HONO and $NO_2$ in cloud and fog to produce $SO_4^{2-}$ in China. Other investigators (Wang et
al., 2016; Ye et al., 2018) have suggested additional chemical pathways for $SO_4^{2-}$ production in China.
Additional research is needed to further understand the chemical pathways for $SO_4^{2-}$ production in
China (Wang et al., 2020b). These pathways are not the focus of this study and, therefore, are not
included in our simulations that leads to the model underpredictions. However, our analysis reveals that
the HONO chemistry and the subsequent production of OH can enhance $SO_4^{2-}$ production in China, so
should be included in air quality models.

The ORI case has mixed performance in simulating observed $NO_3^-$ (Fig. 4). It over-estimates the
daily-averaged observed $NO_3^-$ concentration on some days but captures or under-estimates the observed
concentrations on the other. The over-estimation of winter $NO_3^-$ by CMAQ has been reported in
previous studies (Yu et al., 2005; Appel et al., 2008). Several reactions contribute to the formation of
$HNO_3$ in CMAQv5.3, which then partitions into $NO_3^-$. The heterogeneous hydrolysis of $N_2O_5$ is the
most important night-time reaction, and the oxidation of $NO_2$ by OH is the most important day-time
reaction forming $HNO_3$. CMAQv5.3 uses the parameterization of Davis et al. (2008) for calculating the
uptake coefficient for the heterogeneous hydrolysis of $N_2O_5$. It does not include the organic-coating
effect (Anttila et al., 2006; Riemer et al., 2009) that can lower the uptake coefficient. Several studies
(Brown et al., 2006; Chang et al., 2016; McDuffie et al., 2018; Wang et al., 2020a) have suggested that
the parametrizations used in air quality models, including box model, WRF-CHEM and CMAQv5.3,
produce higher uptake coefficients than that derived from observation-based studies. These higher
uptake coefficients produce high levels of $HNO_3$ and $NO_3^-$ in the model. A recent study also suggests
that the heterogeneous uptake coefficient in China can be even lower than the values derived over the
United States (Wang et al., 2020b). Our current model does not include such lower uptake coefficient
and over-predicts $NO_3^-$ concentrations. Our IRR analysis of the ORI case results suggests that 30.3% of
$NO_3^-$ (averaged over the entire simulation period in Beijing) is produced via night-time heterogeneous
hydrolysis of $N_2O_5$, and 69.7% is produced via day-time oxidation of $NO_2$ by OH. The revised
chemistry further enhances predicted $NO_3^-$ primarily via the enhanced day-time oxidation of $NO_2$.
Overall, night-time heterogeneous hydrolysis of $N_2O_5$ contributes 27.6%, and day-time oxidation of
$NO_2$ contributes 72.4% in the REV case. Consequently, predicted $NO_3^-$ concentrations with the revised
chemistry further are overestimated on most days.

Because of the increased production of $SO_4^{2-}$ and $NO_3^-$, the average concentration of $NH_4^+$ also
increased from 11.1μg/m³ in ORI and to 13.1μg/m³ in REV (Fig. 4). $NH_4^+$ formation is promoted by
enhancing the neutralization of sulfuric acid and $HNO_3$ by ammonia. The dissolution of the precursor
and the ion balance is the main factor for the growth of $NH_4^+$ in CMAQv5.3. The overestimation of
$NO_3^-$ leads to the overestimation of $NH_4^+$ (Liu et al., 2020).






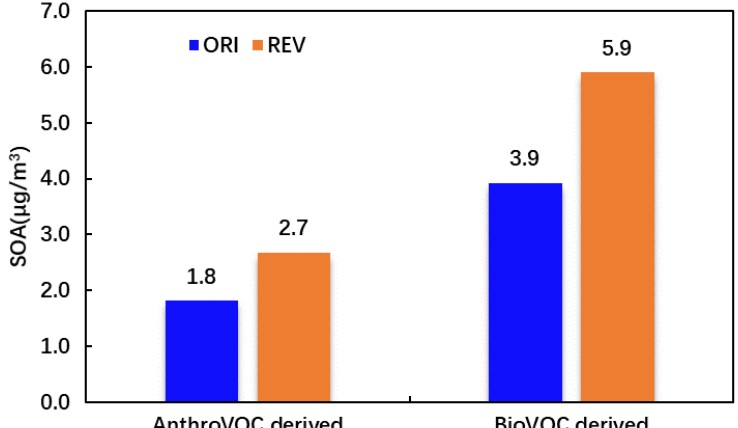


**Fig. 5 Predicted monthly average SOA concentration from anthropogenic VOCs (Anthro-VOC-derived) and biogenic VOCs (Bio-VOC-derived) in Beijing. Numbers in this figure only involve SOA from representative anthropogenic or biogenic VOCs.**


CMAQv5.3 has a comprehensive treatment of organic aerosols (Murphy et al., 2017; Pye et al., 2017; Xu et al., 2018a), including SOA production from anthropogenic-VOC (Anthro-VOC-derived) and biogenic-VOC (Bio-VOC-derived) Fig. 5 displays the Anthro-VOC-derived and Bio-VOC-derived SOA in Beijing.. The REV case enhances the concentration of Anthro-VOC-derived SOA by 0.9 µg/m$^3$ (50%) and Bio -VOC-derived SOA by 2.0 µg/m$^3$ (51%). Enhanced OH from additional HONO enhances the oxidation of VOCs (Table S3) and promotes the SOA formation, which also is reported in previous studies. For example, Xing et al. (2019) used the WRF-CHEM model to examine the impact of HONO chemistry updates on SOA formation over the BTH region in winter and reported that the heterogeneous HONO productions can increase the regional average SOA concentration by 46%. Zhang et al. (2019b) implemented six additional HONO reactions (traffic, soil, biomass burning and indoor emissions, and heterogeneous reactions on aerosol and ground surfaces) in the WRF-CHEM model and reported that it successfully reproduced the observed HONO concentrations in Wangdu. They suggested that the additional HONO reactions can increase 2 to 15 µg/m$^3$ of SOA (meridional-mean) in the BTH region on heavy haze days.

**3.5 Spatial impacts on selected species**

We examine the spatial impacts of the revised HONO chemistry on selected species (HONO, $SO_4^{2-}$, $NO_3^-$, $NH_4^+$, and SOA) in Fig. 6. Predicted average HONO concentrations with ORI are low (<0.18 ppb) over the entire modeling domain. The revised chemistry increases HONO concentrations over the North China Plain (i.e., BTH, Henan, Shandong) by 0.5 to 3.0 ppb. Abundant emissions of $NO_x$ in this area results in higher $NO_2$ concentrations, which subsequently enhance HONO concentrations, as the $NO_2$ reaction on ground is the dominated HONO production source (Fig. 2). It also increases HONO in some other urban areas; however, the impacts in most other areas are relatively small. The ORI case





predicts higher average $SO_4^{2-}$, $NO_3^-$, and $NH_4^+$ concentrations over the North China Plain and the
northeast cites. The revised chemistry enhances average of $SO_4^{2-}$ by 1 to 3 µg/m$^3$, with the maximum
enhancements over the south part of the Hebei province. It increases $NO_3^-$ by up to 1.5 µg/m$^3$ and $NH_4^+$
by up to 1.1 µg/m$^3$ over the North China Plain. It also slightly decreases $NO_3^-$ over the North China
Plain. The revised HONO chemistry decreases $NO_2$ concentration while increasing OH concentration.
Thus, day-time production of $HNO_3$ from the $NO_2$+OH pathway depends on the relative magnitude of
the changes of the reaction rate and tends to increase the production in high-$NO_x$ areas while
decreasing it in low-$NO_x$ areas. $HNO_3$ partitions into $NO_3^-$; thus, changes in $HNO_3$ production leads to
changes in $NO_3^-$ concentration. The ORI case predicts the highest anthropogenic SOA (anthro-SOA)
and biogenic SOA (bio-SOA) concentrations over northeast China and the North China Plain. The
revised model increases anthro-SOA by 0.37 to 1.2 µg/m$^3$ over this area and changes bio-SOA over the
North China Plain and the northeast cities by -2.0 to 2.3 µg/m$^3$. Isoprene emissions in some southern
cities are relatively higher than in cities in North China Plain in the model. Glyoxal and methylglyoxal
generated from isoprene are oxidized by increased OH from the HONO chemistry. SOA derived from
biogenic VOC, therefore, is reduced in some areas in Guangdong.

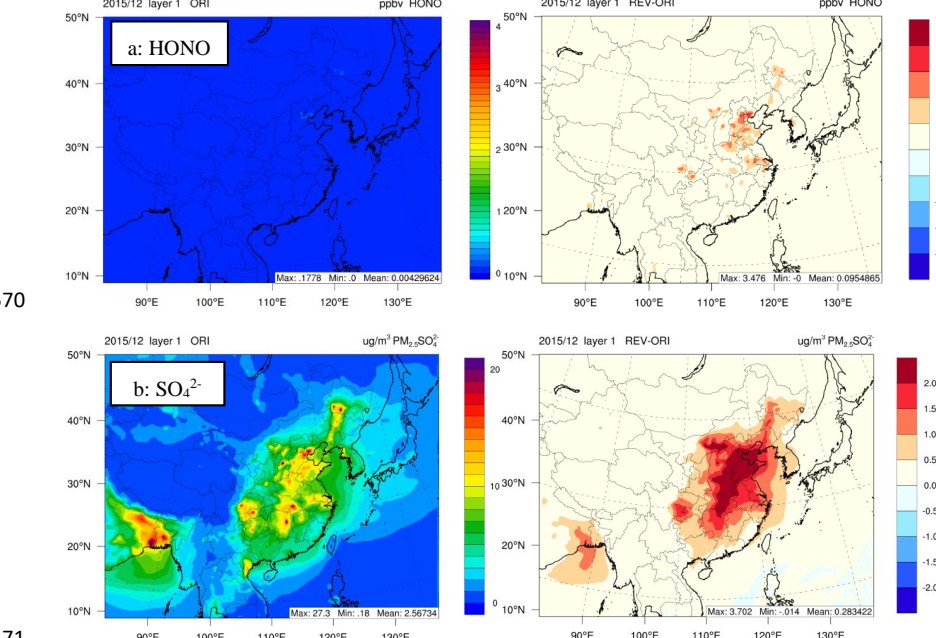








**Fig. 6 Spatial distributions of monthly averaged (a) HONO, (b) sulfate, (c) nitrate, (d) ammonium, (e) anthro-VOC-derived SOA, (f) and bio-VOC-derived SOA concentrations simulated with ORI and the differences (REV-ORI) between the two simulations in December 2015.**



## 4   Summary

The existing HONO chemistry in CMAQv5.3 cannot re-produce the observed winter HONO concentrations in Beijing. Thus, we revised the HONO chemistry in CMAQv5.3 by implementing several heterogeneous HONO formation pathways. Model predictions with the revised chemistry generally agree with observed HONO concentrations, although the model cannot predict the higher observed day-time concentrations. The heterogeneous production on ground accounts for nearly 75% of the total HONO production. Enhanced HONO increases day-time OH concentrations, which also agree well with observed data in Beijing. Predicted OH concentrations with the existing HONO chemistry are lower than observed data almost by a factor of two. The morning OH enhancement rate is lower than the observed rate, and the timing of the peak is delayed. The revised HONO chemistry improves the morning OH enhancement rate and reproduces the daily peak and the timing of the daily peak. Enhanced OH increases the oxidation rates of $SO_2$, $NO_2$, and VOCs in the atmosphere and produces additional secondary pollutants. The revised HONO chemistry moderately enhances $SO_4^{2-}$ concentration in this study. The impact of HONO chemistry on $SO_4^{2-}$ concentration is likely to be greater that shown in this article. For example, HONO chemistry enhances $NO_3^-$, which, in turn, can produce additional $SO_4^{2-}$ via the photolysis of $NO_3^-$ (Zheng et al., 2020). The oxidation of S(IV) by HONO in cloud and fog also can produce additional $SO_4^{2-}$ (Wang et al., 2020). Such pathways are not the focus this study and are not included in the current model. A recent study (Chen et al., 2019) suggests that HONO also can form on snow-covered ground, which can potentially affect wintertime air quality. Thus, a future study incorporating such chemical reactions to comprehensively examine the impact of HONO chemistry on air quality in different seasons and geographical areas is envisioned.

**Acknowledgements**

This work was financially supported by the National Natural Science Foundation of China (41877304, 41907190, 51861135102), and the Youth Innovation Promotion Association, CAS (2018060). This work was also financially and technically supported by Toyota Motor Corporation and Toyota Central Research and Development Laboratories Inc. This work was completed on the "Explorer 100" cluster system of Tsinghua National Laboratory for Information Science and Technology.

**Disclaimer**

The views expressed in this paper are those of the authors and do not necessarily represent the views or policies of the U.S. EPA.

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
