# Peer review of "Improving the representation of HONO chemistry in CMAQ and examining its impact on haze over China"

_Atmospheric Chemistry and Physics, 2021_

## Referee Comment (RC2)

This paper addresses the severe underprediction of nitrous acid (HONO) concentrations by the Community Multi-scale Air Quality model (CMAQ). However, this underprediction is not very surprising because the model omits gas-phase and many heterogeneous reactions that produce HONO. This paper is a welcome addition to the literature on this important topic although much experimental work to better determine these reactions is needed.

The authors provide a summary of the HONO reactions that they include in their version of the gas-phase Carbon Bond mechanism, CB6r3, in Table 1. Although I am doubtful that much, if any, HONO is produced through gas-phase reaction, NO + NO2 + H2O $\rightarrow$ 2 HONO ($k_f$), the authors should check to see if rate coefficients for this reaction and its reverse, HONO + HONO $\rightarrow$ NO + NO2 (+ H2O) ($k_r$) are consistent with the HONO equilibrium constant. The equilibrium constant for this pair of reactions is: Keq = ([HONO] [HONO])/([NO][NO$_2$][H$_2$O]) and Keq = $k_f/k_r$; this expression is correct regardless, if the system is in equilibrium or not.

The value of Keq of 5E-20 derived from Table 1 seems very small considering the value given by Chan et al., (Environ. Sci. Technol., 10, 1976, 674 – 682). [The Chan et al. Keq for HONO was used by Stockwell and Calvert to estimate experimental absorption cross-sections of gas-phase HONO (*J. Photochem., 8*, 193 - 203, 1978) from equilibrium mixtures. The fact that these HONO absorption cross-sections remain consistent today with those produced by more direct methods (see: Burkholder, et al., "Chemical Kinetics and Photochemical Data for Use in Atmospheric Studies, Evaluation No. 19," JPL Publication 19-5, Jet Propulsion Laboratory, Pasadena, 2019 http://jpldataeval.jpl.nasa.gov) support the validity of the Chan et al. Keq for HONO.]

The authors make a surprising statement about HONO chemistry at Lines 69 – 70. They state that the reaction, HO + NO -> HONO, was added to WRF-Chem. But this reaction is included in several of the standard chemical mechanisms in the WRF-Chem model. For example, it is included in the Regional Atmospheric Chemistry Mechanism, version 2 (RACM2).

Lines 148 – 154: The authors correctly state that several heterogeneous HONO producing reactions have been proposed. The possible significance of heterogeneous chemistry for the production of HONO was proposed several decades ago and it would be good if the authors provides some acknowledgement of that fact. For example, Finlayson-Pitts, B. J., and J. N. Pitts Jr. 2000, "Chemistry of the upper and lower atmosphere: Theory, experiments and applications" New York: Academic Press cite a number of investigations of heterogeneous HONO producing reactions. While I acknowledge that the authors' paper is not a historical review, it would be good if they could provide a clear picture of long search by many international researchers for these heterogeneous reactions.

In discussing both gas-phase and aqueous-phase photolysis (Lines 120 – 122; 177 – 184; elsewhere?) The authors make a common mistake in terminology. A photolysis rate is the product of a photolysis frequency (or "photolysis rate coefficient" or "J-value") and the concentration of the substance being photolyzed.  An example of a photolysis rate is J [HONO]. Absorption cross-section and quantum yield data are used for calculating J but it is not a photolysis rate by itself.

The presented measurements and modeling following Line 266 in the Results and Discussions Section are well performed and very interesting. As expected the authors' modeling found that gas-phase chemistry alone can't explain the observed concentrations. It is striking that the HONO day/night behavior and nighttime concentrations in present-day Beijing are

similar to that observed by Platt et al. in Los Angeles during 1980 (Platt et al., Observations of nitrous acid in an urban atmosphere by differential optical absorption, Nature, 285, 312-314, 1980).

In summary, the authors have examined the relative importance of the various HONO producing reactions and shown that HONO can have a dominate effect on the HO budget. These results are potentially relevant to the development of policies to improve air quality in large urban regions. I strongly suggest that the authors address the gas-phase mechanism points as presented in their paper although I doubt modifications of the rate coefficients for the $NO + NO_2 + H_2O \rightarrow 2\ HONO$ and $HONO + HONO \rightarrow NO + NO_2\ (+ H_2O)$ reactions will change their modeling conclusions. Please use either "photolysis frequency" or "photolysis rate coefficient" to describe a J-value to avoid furthering the use of misleading terminology in atmospheric chemistry literature.

---

## Author Response (AR1)

**Point-by-point responses to reviewers**

We thank the two reviewers for the detailed and thoughtful review of our manuscript entitled "Improving the representation of HONO chemistry in CMAQ and examining its impact on haze over China". Those comments are all valuable and very helpful for revising and improving our paper. We think the incorporation of the reviewers' suggestion has led to a much improved manuscript. Detailed below is our response to the issues raised by the reviewers. We also detail the specific changes incorporated in the revised manuscript in response to the reviewers' comments.

**Reviewer #1:**
*Zhang et al. implemented new (heterogeneous) HONO formation mechanisms into the CMAQ model to evaluate HONO formation and impacts in China, especially their Beijing site. The new parameterization shows much better agreement with ground observations in Beijing and the vertical profiles in another field campaign, compared to the original one. In China, in order to get a better prediction of air quality, it is important to have a good HONO parameterization in the model. Some revision should be made before accepting the manuscript.*

*It is critical for the HONO modeling study to clarify why specific parameterization is used. The authors have tried to conduct sensitivity runs and presented results in the SI. However, it is still not convincing why some HONO uptake coefficients were used in the model. Were they based on laboratory experiments, empirical parameters obtained from the field, or simply obtained from other models? These should be clarified.*

[General Comment]: *1.1 For example, at Lines 192-195, are these uptake coefficients based on experimental data? Please clarify here how uncertain they are.*

[Response]:
The selection of uptake coefficients on ground surface and aerosol surface are mainly based on the empirical data derived from either experiments or observations. As the reviewer suggested, we have summarized the variation range of the parameters and several sensitivity results to clarify the associated uncertainties. We referred to some experimental data measured in our laboratory. Experimental data measured on MgO surface fall in the range of $1\text{-}6\times10^{-6}$ as reported by Ma et al. (2017) and on the hematite surface in the range of $1.9\times10^{-7}\text{-}1.6\times10^{-6}$ as reported by Liu et al. (2015) . The derived empirical data obtained by VandenBoer et al.(2013) from the field observation fall in the range of $2\times10^{-6}\text{-}1.6\times10^{-5}$. The empirical uptake coefficient used in models varied widely ranging from $10^{-7}$ to $10^{-3}$ (Table S2). The majority $\gamma_{NO2}$ value employed in literature is about $10^{-6}$. When the uptake coefficient changes by 10 times, the HONO concentration from the heterogeneous reaction on ground surface changes by a factor of two.

Table S2: The uptake coefficient of $NO_2$ used in other studies.

| $\gamma_{NO2}$ | Reference | $\gamma_{NO2}$ | Reference |
|---|---|---|---|
| $1\times10^{-6}$ | (Li et al., 2018a) | $8\times10^{-6}$ | (Liu et al., 2019b) |
| $1\times10^{-5}$ | (Fu et al., 2019) | $1\times10^{-6}$ | (Liu et al., 2014) |
| $1\times10^{-6}$ | (Ndour et al., 2008) | $2\sim7\times10^{-4}$ | (Lu et al., 2018) |
| $1\times10^{-7}$ | (Stemmler et al., 2007) | $5\times10^{-6}$ | (Meng et al., 2020) |
| $10^{-3}\sim10^{-4}$ | (Li et al., 2018b) | $1\sim6\times10^{-7}$ | (Monge et al., 2010) |
| $1\times10^{-6}$ | (Liu et al., 2019a) | $3\times10^{-5}$ | (Spataro et al., 2013) |
| $1\times10^{-6}$ | (Liu et al., 2021) | | |

The detailed revises refer to:

Page 5, Line 170:

The selection criteria and possible ranges of the uptake coefficient are discussed in SI.
Supplemental Information Page 2, Line 47-55:

The selection of uptake coefficients on ground surface and aerosol surface are mainly based on the empirical data derived from either experiments or observations. Experimental data measured on MgO surface fall in the range of $1\text{-}6\times10^{-6}$ as reported by Ma et al. (2017) and on hematite surface in the range of $1.9\times10^{-7}\text{-}1.6\times10^{-6}$ as reported by Liu et al. (2015) . The derived empirical data obtained by VandenBoer et al. (2013) from the field observation fall in the range of $2\times10^{-6}$ $\text{-}1.6\times10^{-5}$. The empirical uptake coefficient used in models varied widely ranging from $10^{-7}$ to $10^{-3}$ (Table S2). The majority $\gamma_{NO2}$ value employed in literature is about $10^{-6}$.

[General Comment]: *1.2 Lines 203-205: Please explain why 1.7/H is used in this study and in previous studies, and how uncertain it is.*

[Response]:

1.7/H represents the ground surface area density (S/Vg) in the model. Effective surface area of ground can be higher than the geometric surface area due to the presence of trees, buildings, and other surface areas. A factor of 1.4-2.2 for the ratio of effective surface area to geometric surface area was measured by Voogt and Oke (1997). Hence, S/Vg =2.2S'/HS'= 2.2/H, S' represents the geometric surface area of the first layer. Previous HONO simulation studies (Vogel et al., 2003; Li et al., 2019; Liu et al., 2019b) used a value of 1.7/H for their modeling studies; we used a value of 1.7 by following these studies. We also perform a sensitivity case by setting S/Vg to 2.2/H. Predicted results are shown in Figure S3. The average HONO increased by 17.2% (from 2.5 ppb with 1.7/H (REV) to 2.9 ppb with 2.2/H).

We have clarified this point in the revised manuscript as follows.

Page 5, Line 179-186:

Following the suggestions of Vogel et al. (2003), Li et al., (2019) and Liu et al., (2019), we use a value of 1.7/H (S/Vg =1.7S'/HS'= 1.7/H, S' represents the geometric surface area of the first layer. 1.7 is the effective surface factor per ground surface in first layer. H is the model's first-layer height.) for surface area-to-volume ratio of ground (S/Vg) to calculate the rate constant for the reaction on ground surfaces. We also conducted sensitivity analysis by using the value of 2.2/H which is suggested from Voogt and Oke

(1997). The result suggests slightly higher concentrations but with similar model performance (details in Figure S4 in Supplemental Information).
Supplemental Information Page 2, Line 103-104:
S/Vg was set to 2.2/H in another sensitivity case. The average HONO increased by 17.2% (from 2.5 ppb with 1.7/H (HONO_REV) to 2.9 ppb with 2.2/H).

[Figure]

Figure S4 A comparison of observed and simulated HONO concentrations in Beijing. HONO observation is denoted as OBS, final simulated HONO concentration with ground surface density of 1.7/H is denoted as REV, and HONO with ground surface density of 2.2/H is denoted as 2.2/H.

[General Comment]: *1.3 Lines 233-236: HNO3 and HCl deposition velocities could be highly uncertain. Please see Jaegle et al. 2018. Please give more details on how HNO3 and HCl deposition velocities were parameterized in the model and how uncertain they are.*

[Response]:

The contribution of HONO from acid displacement (5.5% for $HNO_3$ and 0.7% for HCl) is far less than the heterogeneous reaction on the ground surface (86.2%).The dry deposition velocities of $HNO_3$ and HCl in CMAQ is calculated using a big-leaf resistance model (Wesely, 1989; Wesely, 2007). The total resistance to dry deposition (which is the inverse of v) is calculated as the sum of the bulk surface resistance, $R_{surf}$, the aerodynamic resistance, $R_a$, the quasi-laminar boundary layer resistance, $R_{bc}$. Rsurf includes the influence of vegetation, canopy, ground, etc. Considering the average temperature in our study is around 1.6 ℃ which is above the threshold value for low temperatures as suggested in Jaegle's method (-2℃) (Jaeglé et al., 2018), we used the default mechanism of the surface resistance in CMAQ without modification. However, our model calculated deposition velocities fall within the reported ranges of values (Jaeglé et al., 2018). For example, the modeling value of $v(HNO_3)$ falls within the range of $3 \times 10^{-4}$ cm s$^{-1}$ to 4.1cm s$^{-1}$ with an average velocity of 0.5 cm s$^{-1}$. The

simulated value of v(HCl) falls in $1 \times 10^{-4}$ cm s$^{-1}$ to 0.1 cm s$^{-1}$ with an average velocity of 0.02 cm s$^{-1}$.

$$v = (R_{surf} + Ra + R_{bc})^{-1}$$

$$R_{bc} = \frac{5}{v(\frac{k}{d})^{\frac{2}{3}}}$$

v is the cell friction velocity (m/s); k is kinematic viscosity(cm$^2$/s); d is molecular diffusivity (cm$^2$/s);

We clarified this point in the revised manuscript as follows.

Page 6 Line 211-214:
The dry deposition velocities of HNO$_3$ and HCl in CMAQ is calculated using a big-leaf resistance model (Wesely, 1989; Wesely, 2007). Calculated deposition velocities fall in the reported ranges of values by Jaeglé et al. (2018) (details in Supplemental Information).

[General Comment]: *1.4 Please double check the reference lists. Change it to ACP format. Cite the final ACP papers, not ACPD, e.g. Line 853, Line 977.*

[Response]:
We have updated all references and changed ACPD to the final ACP format.

**Other comments**

[Other Comment]: *1.5 Table 2, 8a and 8b: Please change S/Vg to 1.7/H.*

[Response]:
We have changed the S/V$_g$ to 1.7/H in Table 2 in accordance with the reviewer's comments.

[Other Comment]: *1.6 Line 267: What is "existing heterogeneous hydrolysis of NO2"?*

[Response]:
We have removed "existing" and clarified "heterogeneous hydrolysis of NO2" as follows:
Page 7, Line 242-249:
We performed two different simulations using CMAQv5.3 for December 7-22, 2015. One simulation denoted by "ORI" used the gas-phase HONO chemistry in CB6r3 along with the heterogeneous hydrolysis of NO$_2$ in CMAQv5.3. The implementation of the

heterogeneous hydrolysis of NO₂ in CMAQ has previously been described by Sarwar et al. (2008). They accounted for aerosol surface area as well as the ground surface area provided by leaves and building and other structures. Leaf area was estimated using 2 × LAI/H (LAI is the leaf area index and H is the surface layer height in the model) while building and other structure surface areas were estimated using 0.002 × PURB (PURB = percent urban area of a grid-cell in the model).

[Other Comment]: *1.7: Please show how NMB is calculated here.*

[Response]:
We have added the following text to show how we calculate NMB.
Page 7, Line257-259:
Normalized Mean Bias $(NMB)=100\times\sum(M_i-O_i)/\sum O_i$, $O_i$ is observed HONO concentration, and $M_i$ is the simulated HONO concentration in model (Jaeglé et al., 2018).

[Other Comment]: *1.8 Line 336: What additional sources could that be?*

[Response]:
Aerosol indirect effect (Xing et al., 2017), soil emission (Oswald et al., 2013b), the photolysis of nitrate (Romer et al., 2018) and other unknown sources may cause the underestimation of the daily HONO concentration. We have mentioned these sources separately inL223, L327, L348, L496, SI (L110-L132). In order to avoid repetitive discussion, only the cited sources are added as follows:
Page 8, Line 306-308:
It also increases day-time concentrations, however, predicted values are substantially lower than the observed data, which suggests that additional processes (Oswald et al., 2013a; Xing et al., 2017; Romer et al., 2018) are needed to close the gap between observed and predicted day-time HONO concentrations.

[Other Comment]: *1.9 Fig.1: Please explain what the error bars are.*

[Response]: We have added the following text in line 300 to explain error bars:
Page 9, Line 313:
Error bars represent 5%-95% values of all HONO concentrations.

[Other Comment]: *1.10 Line 365: Please provide values for vehicle exhausts.*

[Response]: We have added the reported values (0.001-0.008) for vehicle exhausts as follows:
Page 10, Line 332:

The observed HONO/NO$_2$ ratios ranging between 0.003 and 0.15 are much higher than reported values in the vehicle exhausts (0.001-0.008) which suggests that HONO formation is governed mainly by the secondary production (Kirchstetter et al., 1996; Kurtenbach et al., 2001).

[Other Comment]: *1.11 Line 464: As shown in Fig. 1, daytime HONO was significantly underestimated in the model. Please discuss how this affects OH concentrations.*

[Response]:
OH concentration is affected not only by the daytime HONO concentration but also by the photolysis rate of HONO. In REV case, we only considered the HONO heterogeneous sources which increase OH concentration as we discussed in section 3.3. Daytime OH concentrations can potentially be higher than the predicted values since daytime HONO concentrations are lower than observed data. However, the aerosol indirect effect may reduce OH concentration as it may slow the HO$_x$ formation rate from HONO. A future study incorporating aerosol indirect effect is needed to improve the representation of HONO chemistry in CMAQ and examining its impact on OH concentration. We revised the text as follows:
Page 14 Line 462-465:
The daytime underestimation of HONO in Fig.1 can potentially lead to the underestimation of OH concentration; however, the aerosol indirect effect may lower the OH concentration by reducing the rates of HO$_x$ formation. Therefore, more accurate HONO simulation needs to consider more complex and significant atmospheric chemical processes.

[Other Comment]: *1.12 Fig. 6: It should show the REV case instead of ORI case, as the REV cases are with HONO updates, the main focus of this study.*

[Response]:
As the reviewer suggested, we have replaced figures in ORI cases to REV cases in the revised manuscript as follows.
Page 20 Fig.6

[Figure]

[Figure]

[Figure]

**Fig. 6 Spatial distributions of monthly averaged (a) HONO, (b) sulfate, (c) nitrate, (d) ammonium, (e) anthro-VOC-derived SOA, (f) and bio-VOC-derived SOA concentrations simulated with REV and the differences (REV-ORI) between the two simulations in December 2015.**

**Reviewer #2:**

[Comment]: *This paper addresses the severe underprediction of nitrous acid (HONO) concentrations by the Community Multi-scale Air Quality model (CMAQ). However, this underprediction is not very surprising because the model omits gas-phase and many heterogeneous reactions that produce HONO. This paper is a welcome addition to the literature on this important topic although much experimental work to better determine these reactions is needed.*

[Response]:
We appreciate the reviewer's recognition for our work and the valuable comments. We have incorporated the reviewer's suggestion into the revised manuscript. Please check the following point-by-point responses.

[Comment]: *2.1The authors provide a summary of the HONO reactions that they include in their version of the gas-phase Carbon Bond mechanism, CB6r3, in Table 1. Although I am doubtful that much, if any, HONO is produced through gas-phase reaction, NO + NO2 + H2O→2 HONO ($k_f$), the authors should check to see if rate*

*coefficients for this reaction and its reverse, HONO + HONO → NO + NO2 (+ H2O) ($k_r$) are consistent with the HONO equilibrium constant. The equilibrium constant for this pair of reactions is: Keq = ([HONO] [HONO])/([NO][NO2][H2O]) and Keq = kf/kr; this expression is correct regardless, if the system is in equilibrium or not. The value of Keq of 5E-20 derived from Table 1 seems very small considering the value given by Chan et al., (Environ. Sci. Technol., 10, 1976, 674 – 682). [The Chan et al. Keq for HONO was used by Stockwell and Calvert to estimate experimental absorption cross-sections of gas-phase HONO (J. Photochem., 8, 193 - 203, 1978) from equilibrium mixtures. The fact that these HONO absorption cross-sections remain consistent today with those produced by more direct methods (see: Burkholder, et al., "Chemical Kinetics and Photochemical Data for Use in Atmospheric Studies, Evaluation No. 19," JPL Publication 19-5, Jet Propulsion Laboratory, Pasadena, 2019 http://jpldataeval.jpl.nasa.gov) support the validity of the Chan et al. Keq for HONO.]*

[Response]:
The rate coefficients of the gas phase decomposition and formation reactions of HONO in this work are used from the CMAQv5.3-CB6 mechanism without any modification (Yarwood et al., 2010). CB6 chemical mechanism has been widely used to simulate many gas-phase species including HONO (Yarwood and Karamchandani–ENVIRON, 2014). The rate constants in CB6 were obtained from the study of Kaiser and Wu (Kaiser and Wu, 1977) who used a Pyrex surface for their experiment. Chan et al (Chan et al., 1976a; Chan et al., 1976b) used a stainless-steel reactor and reported higher rate constants for these reactions. However, the calculated equilibrium constants in both studies are similar ($5 \times 10^{-20}$ in CB6 vs. $6 \times 10^{-20}$ in Chan et al. (1976)).

To clarify this point, we have provided additional discussion in the revised manuscript as follows.

Page 3 Line116-118:

The calculated equilibrium constant in CB6 (Kaiser and Wu, 1977) is similar to reported rate constants by Chan et al ($5 \times 10^{-20}$ in CB6 vs. $6 \times 10^{-20}$ in Chan et al. (Chan et al., 1976a; Chan et al., 1976b).

[Comment]: *2.2 The authors make a surprising statement about HONO chemistry at Lines 69 – 70. They state that the reaction, HO + NO -> HONO, was added to WRF-Chem. But this reaction is included in several of the standard chemical mechanisms in the WRF-Chem model. For example, it is included in the Regional Atmospheric Chemistry Mechanism, version 2 (RACM2).*

[Response]:
Thanks to the reviewer for pointing out the inaccurate expression. The research we cited (Li et al, 2010) did not actually modify the reaction, HO+NO →HONO. This reaction was only taken into consideration in their simulation of HONO formation. We have fixed this error by changing the original text in Page 2 Line 70-74 to "Li et al. (2010) examined the impact of HONO chemistry in Mexico City using the Weather Research and Forecasting model, coupled with chemistry (WRF-CHEM). They considered five different HONO reactions: ① the existing homogeneous reaction between NO (nitric oxide) and OH, ②

the added heterogeneous reaction of $NO_2$ on the aerosol surfaces, ③ the added heterogeneous reaction of $NO_2$ on the ground surfaces, ④ the added heterogeneous reaction of $NO_2$ with semi-volatile organics, and ⑤ the added $NO_2$ reaction with freshly emitted soot.".

[Comment]: *2.3 Lines 148 – 154: The authors correctly state that several heterogeneous HONO producing reactions have been proposed. The possible significance of heterogeneous chemistry for the production of HONO was proposed several decades ago and it would be good if the authors provides some acknowledgement of that fact. For example, Finlayson-Pitts, B. J., and J. N. Pitts Jr. 2000, "Chemistry of the upper and lower atmosphere: Theory, experiments and applications" New York: Academic Press cite a number of investigations of heterogeneous HONO producing reactions. While I acknowledge that the authors' paper is not a historical review, it would be good if they could provide a clear picture of long search by many international researchers for these heterogeneous reactions.*

[Response]:
The heterogeneous formation reactions of HONO have been proposed several decades ago. Finlayson-Pitts reviewed that the heterogeneous reactions of $NO_2$ occurring at the surfaces of aerosol particles, fogs, buildings, and the ground(Finlayson-Pitts, 2000). Subsequent field and experimental studies have reported that HONO can also form from particulate nitrates (Ye et al., 2016; Bao et al., 2018; Romer et al., 2018). The acid displacement reaction can also form HONO on the surface of nitrous acid (VandenBoer et al., 2013). During past 20 years, many modeling studies involving heterogeneous formation of HONO have been conducted The figure below shows the long research history of HONO heterogeneous reactions carried out by many international researchers.

[Figure]

Fig.S1 Research history of heterogeneous reactions of HONO in past decades

As the reviewer suggested, we have added the following discussion in the revised manuscript.
Page 5 Line 152:
The heterogeneous formation of HONO has been studied for several decades. The long research history of HONO heterogeneous reaction can be found in Finlayson-Pitts (Finlayson-Pitts, 2000).

[Comment]: *2.4 In discussing both gas-phase and aqueous-phase photolysis (Lines 120 – 122; 177 – 184; elsewhere?) The authors make a common mistake in terminology. A photolysis rate is the product of a photolysis frequency (or "photolysis rate coefficient" or "J-value") and the concentration of the substance being photolyzed. An example of a photolysis rate is J [HONO]. Absorption cross-section and quantum yield data are used for calculating J but it is not a photolysis rate by itself. Please use either "photolysis frequency" or "photolysis rate coefficient" to describe a J-value to avoid furthering the use of misleading terminology in atmospheric chemistry literature.*

[Response]:
We agree and thank the reviewer to point out our mistake in terminology. We have corrected the term to "photolysis rate coefficient" in the revised manuscript. The corrections are indicated as red.

[Comment]: *2.5 The presented measurements and modeling following Line 266 in the Results and Discussions Section are well performed and very interesting. As expected the authors' modeling found that gas-phase chemistry alone can't explain the observed concentrations. It is striking that the HONO day/night behavior and nighttime concentrations in present-day Beijing are similar to that observed by Platt et al. in Los Angeles during 1980 (Platt et al., Observations of nitrous acid in an urban atmosphere by differential optical absorption, Nature, 285, 312-314, 1980). In summary, the authors have examined the relative importance of the various HONO producing reactions and shown that HONO can have a dominate effect on the HO budget. These results are potentially relevant to the development of policies to improve air quality in large urban regions.*
[Response]:
Thanks for the reviewer's statement about HONO values in present-day Beijing and in Los Angeles during 1980. There are indeed three similar daily characteristics between HONO in Beijing and HONO in Los Angeles during 1980. First of all, from the perspective of concentration, the maximum value of daily HONO concentration in two cities both reached about 2 ppb. Then, HONO has a similar accumulation process at night and a consumption process during the day. Finally, HONO increases the OH concentration through the photochemical reactions during the daytime. When discussing HONO night-time concentration and OH, we referred this influential paper recommended by the reviewer.

Consistent with observations at other cities (Platt et al., 1980; Bernard et al., 2015; Fu et al., 2019), the diurnal variation of observed HONO concentrations in Beijing also reveals higher night-time concentrations than day-time values (Fig. 1b).

HONO can affect greatly the daily OH budget (Platt et al., 1980; Harris et al., 1982; Li et al., 2018c; Lu et al., 2019; Xue et al., 2020).

[Comment]: *2.6 I strongly suggest that the authors address the gas-phase mechanism points as presented in their paper although I doubt modifications of the rate coefficients for the NO + NO2 + H2O → 2 HONO and HONO + HONO → NO + NO2 (+ H2O) reactions will change their modeling conclusions.*

[Response]:

We thank the reviewer for the suggestion. In this study, we used CB6 gas-phase chemical mechanism without any modification. Chemical kinetics in CB6 are based on the results of Kaiser and Wu (Kaiser and Wu, 1977) which are lower than the values reported by Chan et al. (1976a,1976b). We also performed a separate simulation by using the higher rate constants reported by Chan et al. (1976a, 1976b). As expected, the use of higher rate constants did not change predicted HONO concentrations appreciably which reiterates that the contribution of gas-phase chemistry to HONO concentration is relatively small. The average reaction rate of R2 increase from $4\times10^{-6}$ ppb $h^{-1}$ in HONO_ORI to $4\times10^{-4}$ ppb $h^{-1}$ in HONO_Chan. The average reaction rate of R3 increase from $1.7\times10^{-8}$ ppb $h^{-1}$ in HONO_ORI to $1.7\times10^{-6}$ ppb $h^{-1}$ in HONO_Chan. But the reaction of R2 and R3 has manor effect to the total HONO concentration. It is R1 (NO+OH=HONO) dominate the HONO formation in gas phase.

[Figure]

Fig.S5 HONO simulated by default rate coefficients in CB6 (denoted by HONO_ORI) and that measured by Chan et al .(1976) (denoted by HONO_Chan ).

Chemical kinetics of R2 and R3 (Table 1) in CB6 are based on the results of Kaiser and Wu (1977) which are lower than the values reported by Chan et al. (Chan et al., 1976a; Chan et al., 1976b). We also performed a separate simulation by using the higher rate

constants reported by Chan et al. (1976a, 1976b) (Fig. S4). As expected, the use of higher rate constants did not change predicted HONO concentrations appreciably which reiterates that the contribution of gas-phase chemistry to HONO concentration is relatively small.

**[Reference] :**

Bao, F.X., Li, M., Zhang, Y., Chen, C.C., Zhao, J.C., 2018. Photochemical Aging of Beijing Urban PM2.5: HONO Production. Environmental Science & Technology 52, 6309-6316.

Chan, W.H., Nordstrom, R.J., Calvert, J.G., Shaw, J.H., 1976a. Kinetic study of nitrous acid formation and decay reactions in gaseous mixtures of nitrous acid, nitrogen oxide (NO), nitrogen oxide (NO2), water, and nitrogen. Environmental Science & Technology 10, 674-682.

Chan, W.H., Nordstrom, R.J., Galvert, J.G., Shaw, J.H., 1976b. An IRFTS spectroscopic study of the kinetics and the mechanism of the reactions in the gaseous system, HONO, NO, NO2, H2O. Chemical Physics Letters 37, 441-446.

Finlayson-Pitts, B.J., 2000. Chemistry of the upper and lower atmosphere theory, experiments and applications. San Diego, Calif. : Academic Press, San Diego, Calif.

Fu, X., Wang, T., Zhang, L., Li, Q.Y., Wang, Z., Xia, M., Yun, H., Wang, W.H., Yu, C., Yue, D.L., Zhou, Y., Zheng, J.Y., Han, R., 2019. The significant contribution of HONO to secondary pollutants during a severe winter pollution event in southern China. Atmospheric Chemistry and Physics 19, 1-14.

Jaeglé, L., Shah, V., Thornton, J.A., Lopez-Hilfiker, F.D., Lee, B.H., McDuffie, E.E., Fibiger, D., Brown, S.S., Veres, P., Sparks, T.L., Ebben, C.J., Wooldridge, P.J., Kenagy, H.S., Cohen, R.C., Weinheimer, A.J., Campos, T.L., Montzka, D.D., Digangi, J.P., Wolfe, G.M., Hanisco, T., Schroder, J.C., Campuzano-Jost, P., Day, D.A., Jimenez, J.L., Sullivan, A.P., Guo, H., Weber, R.J., 2018. Nitrogen Oxides Emissions, Chemistry, Deposition, and Export Over the Northeast United States During the WINTER Aircraft Campaign. Journal of Geophysical Research: Atmospheres 123, 12,368-312,393.

Kaiser, E.W., Wu, C.H., 1977. A kinetic study of the gas phase formation and decomposition reactions of nitrous acid. The Journal of Physical Chemistry 81, 1701-1706.

Li, D.D., Xue, L.K., Wen, L., Wang, X.F., Chen, T.S., Mellouki, A., Chen, J.M., Wang, W.X., 2018a. Characteristics and sources of nitrous acid in an urban atmosphere of northern China: Results from 1-yr continuous observations. Atmospheric Environment 182, 296-306.

Li, L.J., Hoffmann, M.R., Colussi, A.J., 2018b. Role of Nitrogen Dioxide in the Production of Sulfate during Chinese Haze-Aerosol Episodes. Environmental Science & Technology 52, 2686-2693.

Li, M., Su, H., Li, G., Ma, N., Pöschl, U., Cheng, Y., 2019. Relative importance of gas uptake on aerosol and ground surfaces characterized by equivalent uptake coefficients. Atmospheric Chemistry and Physics 19, 10981-11011.

Liu, J., Li, S., Mekic, M., Jiang, H., Zhou, W., Loisel, G., Song, W., Wang, X., Gligorovski, S., 2019a. Photoenhanced Uptake of NO2 and HONO Formation on Real Urban Grime. Environmental Science & Technology Letters.

Liu, J., Liu, Z., Ma, Z., Yang, S., Yao, D., Zhao, S., Hu, B., Tang, G., Sun, J., Cheng, M., Xu, Z., Wang, Y., 2021. Detailed budget analysis of HONO in Beijing, China: Implication on atmosphere oxidation capacity in polluted megacity. Atmospheric Environment 244.

Liu, Y., Han, C., Ma, J., Bao, X., He, H., 2015. Influence of relative humidity on heterogeneous kinetics of NO2 on kaolin and hematite. Physical Chemistry Chemical Physics 17, 19424-19431.

Liu, Y., Lu, K., Li, X., Dong, H., Tan, Z., Wang, H., Zou, Q., Wu, Y., Zeng, L., Hu, M., Min, K.-E., Kecorius, S.,

Wiedensohler, A., Zhang, Y., 2019b. A Comprehensive Model Test of the HONO Sources Constrained to Field Measurements at Rural North China Plain. Environmental Science & Technology 53, 3517-3525.

Liu, Z., Wang, Y., Costabile, F., Amoroso, A., Zhao, C., Huey, L.G., Stickel, R., Liao, J., Zhu, T., 2014. Evidence of Aerosols as a Media for Rapid Daytime HONO Production over China. Environmental Science & Technology 48, 14386-14391.

Lu, X.C., Wang, Y.H., Li, J.F., Shen, L., Fung, J.C.H., 2018. Evidence of heterogeneous HONO formation from aerosols and the regional photochemical impact of this HONO source. Environ. Res. Lett. 13, 12.

Ma, Q.X., Wang, T., Liu, C., He, H., Wang, Z., Wang, W.H., Liang, Y.T., 2017. SO2 Initiates the Efficient Conversion of NO2 to HONO on MgO Surface. Environmental Science & Technology 51, 3767-3775.

Meng, F., Qin, M., Tang, K., Duan, J., Fang, W., Liang, S., Ye, K., Xie, P., Sun, Y., Xie, C., Ye, C., Fu, P., Liu, J., Liu, W., 2020. High-resolution vertical distribution and sources of HONO and NO$_2$ in the nocturnal boundary layer in urban Beijing, China. Atmospheric Chemistry and Physics 20, 5071-5092.

Monge, M., D'Anna, B., Mazri, L., Giroir-Fendler, A., Ammann, M., Donaldson, D.J., George, C., 2010. Light changes the atmospheric reactivity of soot. Proceedings of the National Academy of Sciences of the United States of America 107, 6605-6609.

Ndour, M., D'Anna, B., George, C., Ka, O., Balkanski, Y., Kleffmann, J., Stemmler, K., Ammann, M., 2008. Photoenhanced uptake of NO2 on mineral dust: Laboratory experiments and model simulations. Geophysical Research Letters 35.

Oswald, R., Behrendt, T., Ermel, M., Wu, D., Su, H., Cheng, Y., Breuninger, C., Moravek, A., Mougin, E., Delon, C., 2013a. HONO emissions from soil bacteria as a major source of atmospheric reactive nitrogen. Science 341, 1233-1235.

Oswald, R., Behrendt, T., Ermel, M., Wu, D., Su, H., Cheng, Y., Breuninger, C., Moravek, A., Mougin, E., Delon, C., Loubet, B., Pommerening-Röser, A., Sörgel, M., Pöschl, U., Hoffmann, T., Andreae, M.O., Meixner, F.X., Trebs, I., 2013b. HONO Emissions from Soil Bacteria as a Major Source of Atmospheric Reactive Nitrogen. Science 341, 1233-1235.

Romer, P.S., Wooldridge, P.J., Crounse, J.D., Kim, M.J., Wennberg, P.O., Dibb, J.E., Scheuer, E., Blake, D.R., Meinardi, S., Brosius, A.L., Thames, A.B., Miller, D.O., Brune, W.H., Hall, S.R., Ryerson, T.B., Cohen, R.C., 2018. Constraints on Aerosol Nitrate Photolysis as a Potential Source of HONO and NOx. Environmental Science & Technology 52, 13738-13746.

Spataro, F., Ianniello, A., Esposito, G., Allegrini, I., Zhu, T., Hu, M., 2013. Occurrence of atmospheric nitrous acid in the urban area of Beijing (China). Sci. Total Environ. 447, 210-224.

Stemmler, K., Ndour, M., Elshorbany, Y., Kleffmann, J., D'Anna, B., George, C., Bohn, B., Ammann, M., 2007. Light induced conversion of nitrogen dioxide into nitrous acid on submicron humic acid aerosol. Atmos. Chem. Phys. 7, 4237-4248.

VandenBoer, T.C., Brown, S.S., Murphy, J.G., Keene, W.C., Young, C.J., Pszenny, A.A.P., Kim, S., Warneke, C., de Gouw, J.A., Maben, J.R., Wagner, N.L., Riedel, T.P., Thornton, J.A., Wolfe, D.E., Dubé, W.P., Öztürk, F., Brock, C.A., Grossberg, N., Lefer, B., Lerner, B., Middlebrook, A.M., Roberts, J.M., 2013. Understanding the role of the ground surface in HONO vertical structure: High resolution vertical profiles during NACHTT-11. Journal of Geophysical Research: Atmospheres 118, 10,155-110,171.

Vogel, B., Vogel, H., Kleffmann, J., Kurtenbach, R., 2003. Measured and simulated vertical profiles of nitrous acid—Part II. Model simulations and indications for a photolytic source. Atmospheric Environment 37, 2957-2966.

Voogt, J.A., Oke, T.R., 1997. Complete Urban Surface Temperatures. Journal of Applied Meteorology 36,

1117-1132.

Wesely, M., 2007. Parameterization of surface resistances to gaseous dry deposition in regional-scale numerical models. Atmospheric Environment 41, 52-63.

Wesely, M.L., 1989. Parameterization of surface resistances to gaseous dry deposition in regional-scale numerical models. Atmospheric Environment (1967) 23, 1293-1304.

Xing, J., Wang, J., Mathur, R., Wang, S., Sarwar, G., Pleim, J., Hogrefe, C., Zhang, Y., Jiang, J., Wong, D.C., Hao, J., 2017. Impacts of aerosol direct effects on tropospheric ozone through changes in atmospheric dynamics and photolysis rates. Atmospheric Chemistry and Physics 17, 9869-9883.

Yarwood, G., Karamchandani–ENVIRON, P., 2014. IMPLEMENTATION AND EVALUATION OF NEW HONO MECHANISMS IN A 3-D CHEMICAL TRANSPORT MODEL FOR SPRING 2009 IN HOUSTON FINAL REPORT.

Yarwood, G., Whitten, G.Z., Jung, J., 2010. Development, Evaluation and Testing of Version 6 of the Carbon Bond  Chemical Mechanism (CB6)

Ye, C., Gao, H., Zhang, N., Zhou, X., 2016. Photolysis of Nitric Acid and Nitrate on Natural and Artificial Surfaces. Environmental Science & Technology 50, 3530-3536.